# Recent Fabrication Methods to Produce Polymer-Based Drug Delivery Matrices (Experimental and In Silico Approaches)

**DOI:** 10.3390/pharmaceutics14040872

**Published:** 2022-04-15

**Authors:** Anna Procopio, Elena Lagreca, Rezvan Jamaledin, Sara La Manna, Brunella Corrado, Concetta Di Natale, Valentina Onesto

**Affiliations:** 1Biomechatronics Laboratory, Department of Experimental and Clinical Medicine, University Magna Graecia of Catanzaro, 88100 Catanzaro, Italy; anna.procopio@unicz.it; 2Department of Chemical, Materials & Industrial Production Engineering, University of Naples Federico II, 80131 Naples, Italy; elena.lagreca@iit.it (E.L.); jamaledinrezvan@gmail.com (R.J.); 3Istituto Italiano di Tecnologia, Largo Barsanti e Matteucci 53, 80125 Naples, Italy; 4Department of Pharmacy, University of Naples Federico II, 80131 Naples, Italy; sara.lamanna@unina.it; 5Interdisciplinary Research Centre on Biomaterials (CRIB), University of Naples Federico II, 80131 Naples, Italy; brunella.corrado@iit.it; 6Institute of Nanotechnology, National Research Council (CNR-Nanotec), Campus Ecotekne, Via Monteroni, 73100 Lecce, Italy

**Keywords:** polymers, micro and nano encapsulation, microfluidics, microneedles, in silico approaches

## Abstract

The study of novel drug delivery systems represents one of the frontiers of the biomedical research area. Multi-disciplinary scientific approaches combining traditional or engineered technologies are used to provide major advances in improving drug bioavailability, rate of release, cell/tissue specificity and therapeutic index. Biodegradable and bio-absorbable polymers are usually the building blocks of these systems, and their copolymers are employed to create delivery components. For example, poly (lactic acid) or poly (glycolic acid) are often used as bricks for the production drug-based delivery systems as polymeric microparticles (MPs) or micron-scale needles. To avoid time-consuming empirical approaches for the optimization of these formulations, in silico-supported models have been developed. These methods can predict and tune the release of different drugs starting from designed combinations. Starting from these considerations, this review has the aim of investigating recent approaches to the production of polymeric carriers and the combination of in silico and experimental methods as promising platforms in the biomedical field.

## 1. Introduction

Drug delivery systems are applied to transfer drugs and vaccines to a targeted site within the body with a controlled release to increase their therapeutic efficacy. Such goals are firstly reached by ameliorating solubility and chemical stability of the therapeutic agents, and secondly by enhancing their pharmacological activity, and reducing adverse effects, providing, and preserving the concentrations of therapeutic agents at the targeted site. In the plethora of drug delivery devices, biodegradable polymeric-based systems in the micro- and nanometer range (emulsions, particles, needles, and fibers) became very popular thanks to their unique properties, such as biocompatibility, biodegradability, low-cost materials, easy scalability, and surface functionalization. In addition, they are able to improve drug stability and bioavailability and realize controlled release or active targeting drug delivery [1]. Micro- and nanoemulsions are well-diffused systems for oral and for injectable delivery of molecules, and among these, oil-in-water nanoemulsions became quite attractive for the encapsulation of lipophilic molecules and for the enhancement of drug stability and delivery [2]. Equally, micro- or nanoparticles have drawn great attention [3]. Indeed, since translocating drugs into the cytoplasm of targeted cells is a major challenge in the treatment of various diseases [4,5], drugs started to be encapsulated into either micro- , nano-, or both, structured systems, paving the way to an enhanced cytoplasmic delivery, decreasing drugs plasma degradation [6]. Moreover, due to their simple fabrication processes, polymeric microparticles represent promising devices for industrial scale-up and the possibility to realize a controlled drug release [7]. In detail, a controlled drug release from micro/nano particles can be reached by modulating monomer composition, crystallinity, glass transition temperature (*T*_g_), porosity, and inherent viscosity. In recent years, different kinds of polymer were investigated, because of non-toxic metabolites that were easily eliminated by excretory organs, such as liver or kidney, for the production of microparticles as alginate [8,9], dextran [10], chitosan [11], poly(ethylene glycol) diacrylate (PEGDA) [12] and poly-(lactic-co-glycolic acid) (PLGA) [13]. Due to PLGA properties, PLGA microparticles became a very popular drug delivery system for different administration routes, such as the pulmonary, oral, and transdermal, for both lipophilic and hydrophilic drugs [7]. In addition to oral delivery, transdermal routes have been proposed to improve PLGA MPs penetration into the stratum corneum [7]. In this scenario, the use of microneedles has emerged as a promising tool to perforate the outermost of the skin and allow drug translocation into the dermis [14]. In detail, microneedles are sub-millimetric structures, which can be painlessly inserted into the skin, where they can release the encapsulated molecules upon degradation or dissolution of the polymeric matrix. They can be divided into several categories, i.e., dissolvable, degradable, hydrogel-forming, or coated [15]. Recently, stimuli-responsive polymeric microneedles have been developed to facilitate and monitor the release of drugs [16]. Inspired by nature, researchers have introduced a new category of microneedle patches that are known as biomimetic or bioinspired microneedles which aim to ameliorate the penetration ability [17].

In the last years, the encapsulation of drugs into electrospun fibrous matrices represented an efficient solution for the in situ delivery, owing to the biocompatibility, flexibility, and easy and cost-effective fabrication of the polymeric fibers [18,19]. Electrospun fibers can incorporate active agents within the lumen thanks to their porous nature or biomolecules can be immobilized on the surface of the fibers through either chemical immobilization or physical adsorption [20]. Porosity, structure, morphology of the fibers, and consequently the drug release, can be controlled through electrospinning solution variables (selected polymers and solvents, conductivity, and viscosity), process parameters (tip of the needle-collector distance, flow rate, high voltage, and geometry of the collector), and environmental conditions (humidity and temperature) [20].

Together with experimental parts, mathematical models have been developed to optimize and tune in vitro and in vivo drugs release [21]. In particular, in silico models interpret the physical and chemical mechanisms of the release, thereby decreasing the number of experiments [21,22,23].

The present review focuses on emerging strategies for drug delivery, such as polymeric particles, oil-in-water layered nanoemulsions, electrospun fibers, and micro/nanoneedles obtained by different techniques. At the same time, we explain the importance of the in silico approach as an outstanding method to predict and tune the release profile. Moreover, the fabrication methods, advantages, and disadvantages involving each system will be covered, and at the end, marketing and future perspectives will be discussed.

## 2. Micro- and Nano-Technologies

### 2.1. Micro- and Nano-Particles

Polymeric nanoparticles and microparticles have attracted great attention as outstanding delivery tools to boost the bioavailability and bio-distribution of the drugs [24]. As previously reported, PLGA, chitosan, or alginate are the most popular polymers to obtain a controlled drug delivery. Alginate microparticles have been extensively used to enhance probiotic stability during the GI transit. Di Natale et al., proposed a simple method for the encapsulation of the probiotic Lactobacillus paracasei CBA L74(*L. paracasei CBA L74*) in sodium alginate microparticles obtained by the water-in-oil emulsion technique with a fine evaluation of cross-link range by microrheology analysis to better estimate the useful condition of probiotic encapsulation [8]. On the other hand, PLGA microparticles (MPs) are the most common drug delivery system for hydrophilic and lipophilic drugs due to the simple preparation procedure and the possibility of achieving a controlled drug release by modifying its bio-degradation rate, acting on the polymer chemistry, such as initial MW, stereochemistry (composition in D and L), glycoside units content, or either end-group functionalization, microparticles porosity, or in combination [7]. Furthermore, a controlled drug release can be designed by selecting the encapsulation strategy that is involved on the MP internal structure and on the drug distribution within the MPs. A simple predictive in silico-supported approach has been proposed for the optimization of the pharmacokinetics of an MP-based formulation to predict the release of curcumin (CUR) from a designed combination of different PLGA MPs types. This model can be easily adapted for other molecules, including hydrophilic molecules, and it will be helpful to release drugs with a precise timing and amount, increasing the therapeutic efficiency and thus reducing the side effects [21]. PLGA MPs are also well employed for protein encapsulations, such as collagenase [25], vascular endothelial growth factor (VEGF) [26], and laccase [27].

Nanoparticles refer to both reservoir-based systems (nanocapsules) and matrix-based systems (nanospheres), which can be recognized by their morphological characteristics. Nanocapsules include an oily core containing the drug, bordered by a polymeric shell that can control the release profile of the drug. Nanospheres consist of a continuous polymeric network in which the drug can be kept inside or adsorbed onto the surface. These two types of polymeric NPs are depicted in Figure 1. Polymeric NPs are a promising tool for enhancing efficacy and delivery of poorly water-soluble drugs by increasing solubility and permeability, protecting drugs from degradation, prolonging their residence time in the gastrointestinal tract, protecting mucosa from the toxic effects of drugs, and decreasing their toxicity by limiting nonspecific uptake [28]. Due to their nanoscale dimensions, NPs are able to be internalized by the cells and to deliver the drug to the target site on either a nonspecific (through enhanced permeability and retention (EPR)) or specific manner (by binding specific target receptors).

Nowadays, researchers are focusing on finding new and different methods of NP production and modification, in order to obtain a fine control on polymer degradation, drug release, and drug stability. In addition, muco-adhesion properties of nanoparticles are also being improved by using muco-adhesive polymers (as cationic polymers, anionic polymers, or thioled polymers) [29]. Thanks to their muco-adhesive properties, nanoparticles acquire: (i) prolonged residence time, which enhances absorption; (ii) first pass metabolism inhibition and thus a drug bioavailability improvement; and (iii) drug degradation avoidance by preserving them from the cell environment [29]. Another interesting aspect is the preparation of highly engineered NPs able to realize a specific targeting to reduce the side effects and toxicity of encapsulated drugs. There is great attention to validate new and different targeting strategies including, e.g., active targeting moieties or biomimetic coating by covering it with an outer surface mimicking the composition and functionality of the cell’s external membrane (i.e., PEG [30] or hyaluronic acid coating [31]) to improve the retention of NPs at the target site as well as improve and control the drug release. PEGylation is one of the most exploited strategies to obtain a biomimetic nanocarrier. PEG is a biocompatible polymer which when immobilized onto surfaces confers protein and cell resistance [32,33]. Recently, PEGylation has been replaced by cancer cell membrane coating [34] which can deliver therapeutics directly to tumor cells thanks to its ability to join biomimetic features with homotypic targeting phenomena [35]. Homotypic targeting strategy due to the presence of cancer cell adhesion molecules as membrane receptors including cadherins, selectin and integrins, immunoglobulin superfamily (Ig-SF), and lymphocyte-homing receptors (e.g., CD44) [36] was exploited for targeting different tumor cells, such as melanoma, breast cancer, as well as to address metastasis [37].

In the following paragraph, several techniques to produce nano and microparticles will be discussed.

### 2.2. Polymeric Layered Secondary Oil-in-Water Nanoemulsions

Nanoemulsions are dispersions of very small droplets of one immiscible liquid in another immiscible liquid. They are colloidal systems characterized by small particles that avoid the precipitation and coalescence phenomena thanks to the Brownian motion of the droplets that counteracts the kinetic instability caused by gravity or viscosity [38]. Their fabrication involves the use of mechanical devices, such as high-pressure valve homogenization, microfluidization, and sonication [38,39]. The two immiscible phases involved in the emulsion assembly are called oil and water and nanoemulsions are typically classified into oil-in-water (O/W) and water-in-oil (W/O) types. Typically, the two fluids are oil and water, but other immiscible liquids can also be water emulsified as unsaturated fatty acids (oleic acid, linolenic acid, linoleic acid) or saturated fatty acids (lauric acid, capric acid, myristic acid) and fatty acid ester as [40]. O/W nanoemulsions consist of small oil droplets dispersed in an aqueous phase, while W/O nanoemulsions consist of small water drops dispersed in an oil phase. O/W nanoemulsions are much more diffused than W/O ones, because of their interesting physicochemical and functional properties, such as high loading efficiency for lipophilic molecules and high bioavailability. Nanoemulsions are well established systems for the encapsulation of vitamins, natural antimicrobials, nutraceuticals, and cannabinoids [41,42]. Nonetheless, they present some drawbacks, such as a short shelf life even though stabilized by the use of emulsifiers: indeed, they are vulnerable to destabilization mechanisms, such as creaming, sedimentation, coalescence, flocculation, and Ostwald ripening [43] In order to improve stability and shelf life multilayered nanoemulsions were introduced. In this regard, McClements and his group proposed to produce secondary emulsions obtained with a polyelectrolyte thin layer, adsorbed exploiting the electrostatic interaction with an ionic emulsifier of opposite charge on the oil droplets surface to improve emulsion stability. However, such a method did not completely resolve the stability issue, for that reason the same author suggested using a high-pressure homogenizer to process the emulsion together with a food grade polymer. However, this upgrade did not produce a stable colloidal system since the final nanoemulsions had a wide size distribution, predictably leading to fast destabilization. In 2014, Vecchione et al. investigated molecular factors contributing to the stability and interfacial properties of nanoemulsions and developed a tunable new post-processing method based on multi-redispersion at high pressure that allowed improving of secondary emulsion homogeneity and polymer coating density, monodispersion, and stability over time [43]. Among biodegradable polymeric and polycationic materials, the most exploited is chitosan and its derivatives (e.g., thiolated chitosan or glycol chitosan). Chitosan layered nanoemulsions are well established oral delivery systems for lipophilic molecules (Coenzyme Q10 [44,45], lycopene [46], curcumin [47,48,49]) or essential oils. Chitosan nanoemulsions also demonstrated having muco-adhesion properties [50] that could be improved with thiolated chitosan derivatives [49,51,52]. Vecchione et al. developed engineered chitosan layered nanoemulsions in which peptides achieving active or passive targeting or [31], stimuli response systems [53,54]. Recently a new kind of biomimetic layered nanoemulsion, namely, CM-NEsoSomes was proposed with an external coating of cancer cell membrane to increase biomimetic features and homotypic targeting properties [55]. This approach was based on electrostatic interactions between negatively charged cell membrane and positively charged secondary nanoemulsions. These new layered nanoemulsions represent a useful biocompatible system for the treatment and diagnosis of several human diseases, including cancer, as well as for vaccination and prevention from viral infection. For all these recent progresses, layered nanoemulsions caught the attention of both the pharma and food industries for different administration routes, such as oral delivery, transdermal delivery.

Layered nano-emulsions can be classified as polymer-lipid hybrid nanoparticles (PLN) due to their double nature polymeric coatings and lipid oil core. The main advantage of PLN is the combination of properties of lipid core-based and polymeric-based nanosystems [56]. The starting building blocks of PLN are generally regarded as safe (GRAS) materials that are either already used in human pharmaceuticals approved by FDA or known biocompatible polymers (such as poly(ethylene glycol) and chitosan) and lipids (oil, phospholipids, glycerol). PLN can generally be obtained by a fast and simple procedure and as previously claimed for nanoemulsions, these systems have high capability of co-encapsulating therapeutic and imaging agents of different properties. For all these recent progresses, PLN caught the attention of both the pharma and food industries for different administration routes, such oral delivery and transdermal delivery [56,57]. All these characteristics make PLN one of the most diffused nanocarrier types for the development of highly modulable and biocompatible nanodelivery systems.

In the following paragraph several techniques for the production of nano and microparticles will be discussed.

### 2.3. Different Methods for Polymeric Micro and Nanoparticles Synthesis

#### 2.3.1. Nanoemulsion Assembly

As described in Section 2.2, polymeric layered secondary oil-in-water nanoemulsions are defined as dispersions of immiscible liquid phases where one of the liquids is distributed in the other in the form of small drops. The principal components of nanoemulsions are:oil phase (as corn oil, soybean oil, olive oil, peanut oil, and flaxseed oil with short or medium chain length, which are harmless to the human body) [58]water phase (generally water)surfactants (emulsifier agents, such as non-ionic surfactant, e.g., lecithin, poloxamers, or non-electrolytes, such as glycerol or xylitol [59] or food-grade emulsifiers, such as β-lactoglobulin, whey protein isolate and octenyl succinic acid modified starch, gum Arabic acid and other polysaccharides) [58]co-surfactants (short chain alcohol, organic ammonia, single and double amino acid glyceride and the most used is short chain alcohol, commonly used to aid the emulsifier has ethanol, ethylene glycol, propylene glycol, propylene triol, and poly glyceride) [60].

Different methods are available to prepare nanoemulsions, and their choice is crucial for obtaining high stability and the desired final properties. Nowadays, nanoemulsions can be produced by high and low emulsification energy methods. High-energy methodologies employ mechanical devices, such as micro fluidizers, high pressure homogenizers, microfluidics, high-energy stirring, membrane emulsification, or ultrasonication, that produce strong powerful disrupting forces able to create very small droplets [61].

On the other side, low-energy approaches are based on complex interfacial hydrodynamic phenomena, and these methods are subject to the system composition properties. The most diffused low-energy methods are solvent displacement, spontaneous emulsification, and phase inversion including phase inversion temperature method and the emulsion inversion point method. In more detail, in low energy systems, an initial water-in-oil macroemulsion is converted into an oil-in-water nanoemulsion by altering the composition or temperature [62]. Another interesting approach is the combination of high-energy and low-energy emulsifications [63], by which it is possible to assemble a reverse nanoemulsion into highly viscous systems. As previously reported, nanoemulsion stability can be improved with the deposition of a polymeric layer. This coating can be easily obtained through layer-by-layer (LbL) approaches, which involve the sequential deposition of polymeric layers of opposite charge exploiting the electrostatic interaction between them [44,45,46]. The starting building block is an oil-in-water emulsion containing charged emulsifier-coated oil droplets usually obtained by high-energy methodologies. This charged emulsion is then mixed with a solution containing oppositely charged biopolymers to allow the adsorption to the oil droplet surfaces and formation of an additional coating [44,45,46]. The success of layer deposition can be easily controlled by measuring the increase in particle size and the change in zeta-potential after each successive layer is deposited. By repeating these steps numerous times, it is possible to obtain multilayer-coated oil droplets. One of the greatest adversities encountered in the deposition process is finding the optimal polymer concentration able to fully coat the droplets surfaces avoiding flocculation or destabilization mechanism related to an excess or a lack of polymer concentration. The right concentration can be obtained by monitoring zeta-potential and size values or by fluorescence, or with more sophisticated techniques, such as isothermal titration calorimetry [44,45,46].

However, multilayer emulsions formation and stability are strictly influenced by the combination of biopolymers used and by the employed solution conditions, such as pH and ionic strength.

#### 2.3.2. Emulsion Solvent Evaporation Method

In emulsion solvent evaporation, the polymer is dissolved in an organic phase and the active ingredient (e.g., drug) is added by dissolution or dispersion into the polymeric solution. A water phase (e.g., polyvinyl acetate; PVA), is added to emulsify the solution, by using high-speed homogenization or ultrasonication, resulting in a dispersion of nanodroplets. A suspension of NPs is formed by evaporation of the polymer-solvent, which is allowed to diffuse through the continuous phase of the emulsion. The solvent is eliminated either by a continuous magnetic stirrer at ambient temperature or in a slow process of reduced pressure. After evaporation of the solvent, the solidified NPs can be washed and collected by centrifuge, followed by a freeze-drying process for long-term storage. This method is applied for the fabrication of nano-spheres and microparticles.

#### 2.3.3. Single Emulsion Technique

Hydrophobic drugs are completely dissolvable in organic solvents and thereby poorly soluble in water. One of the most common methods to encapsulate such drugs is by the oil-in-water (O/W) emulsion/solvent evaporation technique. Briefly, this technique involves the generation of the first emulsion (O/W) in which polymer and drug are dissolved completely in an appropriate solvent organic, such as dichloromethane. The first emulsion is added by simple agitation, sonication, or homogenization to the water phase. Subsequently, the evaporation of the emulsion internal phase by stirring takes place. Serval examples of micro and nanoparticles obtained with the O/W methodology for lipophilic drugs, such as donepezil [64], noscapine [65] or steroids are reported. Gonzales et al. developed poly(lactic-co-glycolic acid) nanoparticles containing turmeric compounds (turmeric-PLGA-NPs) using emulsion-solvent evaporation demonstrating the ability of producing NPs in the range of 145 nm with a low polydispersity index (PdI) in size distribution and zeta potential of −21.8 mV. Moreover, this formulation showed a good encapsulation efficacy percentage (about 72%) and a good time stability over a period of one month. In vitro study displayed a release of curcumin ruled by diffusion and relaxation of the polymeric matrix [66].

#### 2.3.4. Double Emulsion

The double emulsion method is a widely used technique to encapsulate hydrophobic or hydrophilic drugs. These complex systems consist of a so-called “emulsion of the emulsion”, in which droplets of the dispersed phase contain one or more droplets themselves. Depending on the type of drug to be loaded, there are two categories of double emulsion systems: water-in-oil-in-water (W/O/W) emulsions, where water droplets are dispersed in oil droplets, which in turn are dispersed in a continuous aqueous phase, and O/W/O emulsions where oil droplets are located within water droplets that are dispersed within a continuous oil [7]. Generally, in order to stabilize the emulsion, a lipophilic emulsifier is contained in the oil phase while a hydrophilic emulsifier is mixed with the outer aqueous phase [67]. Compared to the single emulsion method, W/O/W allows a more efficient and stable encapsulation of water soluble drugs or hydrophilic molecules, protecting the internal active ingredient from the external agent (e.g., light, enzymatic degradation, oxidation) [21]. Moreover, the release of active internal ingredients is more controlled since the presence of the middle oil layer act as a liquid membrane. Typically, proteins and peptides are successfully encapsulated with this technique [7,26,27,68,69]. Biodegradable polymers, such as PLGA, are commonly used in drug delivery systems for encapsulation since the products of their degradation are non-toxic and biocompatible.

#### 2.3.5. Emulsification/Solvent Diffusion

In the emulsification/solvent diffusion technique, as a modified version of the solvent evaporation, the water-soluble solvent for instance is acetone or methanol along with the water-insoluble organic solvents, such as dichloromethane or chloroform, as an oil phase are applied. Due to the spontaneous diffusion of water-soluble solvent, interfacial turbulence between two phases is generated and thereby forms the smaller particles. When the concentration of water-soluble solvent rises, a noticeable decrease in particle size is observed. The solvent diffusion allowed for easy control over the dimension and morphology of particles during the solidification of polymer within the droplets. This technique was highly diffused with modifications as membrane emulsification/solvent diffusion processes [70] or high pressure emulsification-solvent evaporation [71]. The application of microporous membranes allowed for the real-time particle size and size distribution control during the emulsification step. In particular, Imbrogno et al. produced polycaprolactone microparticles with mean diameter of 20 μm, spherical shape, and smooth surface [70]. Ueda et al. used high pressure emulsification-solvent evaporation to optimize the preparation of the loperamide hydrochloride (LPM)-loaded poly(DL-lactic acid) (PLA) nanoparticles—the organic phase evaporation under reduced pressure with the addition of ethanol in the organic phase improved the drug entrapment due to the rapid polymer precipitation [71]. The emulsion-solvent evaporation technique is usually used for loading drugs in polymer nanoparticles because it is an adaptable process, allowing for many drug–polymer pair combinations. Recently, the anti-epileptic drug, lamotrigine, was loaded into biodegradable polymer poly-ε-(d,l-lactide-co-caprolactone) (PLCL) nanoparticles stabilized using various molecular weights and concentrations of polyvinyl alcohol by emulsification solvent evaporation technique. Results indicated that varying polyvinyl alcohol concentration and molecular weight used to stabilize PLCL nanoparticles also confers versatile properties for designing nanoparticles with wide range of properties. In detail, particle size, entrapment efficiency, drug loading capacity, yield percentage, and zeta potential values for the PLCL nanoparticles decreased with increasing PVA concentration. However, above a critical concentration of drug, which depends on the drug and polymer, nanoparticles tend to precipitate. Niyom et al. investigated the role of drug solubility and miscibility in the polymer phase to select the optimal polymeric matrix. In detail, three poorly soluble drugs (ibuprofen, naproxen methyl ester, and naproxen) were selected as model molecules and encapsulated with various loadings in polycaprolactone nanoparticles by the emulsion-solvent evaporation method. They identified two principle parameters to evaluate drugs miscibility into the polymer matrix: Flory–Huggins interaction parameter and Hansen partial solubility parameter [72]. The Flory–Huggins (F–H) solubility equation has been extensively used to define the solubility of a small-molecule drug in a polymeric matrix and thus to design a stable amorphous solid dispersion. F–H parameters (*χ*) (Equation (1)) describe the thermodynamic features of drug–polymer solutions by considering the enthalpic and entropic contributions in solubility. F–H parameters (*χ*) can be calculated with the following equation:(1)$$\left(\frac{1}{T_{m}^{mix}}-\frac{1}{T_{m}^{pure}}\right)=\frac{R}{\Delta H_{m}}\left[\ln\phi_{drug}+\left(1-\frac{1}{m}\right)\phi_{polymer}+\chi \phi_{polymer}^{2}\right]$$
where $T_{m}^{mix}$ is the melting temperature of the drug in the presence of the polymer, $T_{m}^{pure}$ and Δ*H_m_* are the melting temperature and enthalpy of fusion of pure drug, respectively, *m* is the molar volume ratio of the polymer to drug, *ϕ_drug_* and *ϕ_polymer_* are the volume fractions of drug and polymer, and *R* is the gas constant [72]. Thermal properties ($T_{m}^{mix},T_{m}^{pure}$ and Δ*H_m_*) of the polymer/drug blends and drugs can be determined from the first heating curves of differential scanning DSC measurements. Specifically, the solubility between drugs and polymer increased with a decrease of Flory–Huggins interaction values: for *χ* > 0.5 values the drug molecules in the mixture tend to separate [52]. The Hansen solubility parameter (*δ_t_*) employed a group contribution method to approximately calculate the interaction parameter. The *δ_t_* can be calculated using the group contribution method of Van Krevelen and Hoftyzer:(2)$$\delta_{t}^{2}=\delta_{d}^{2}+\delta_{p}^{2}+\delta_{h}^{2},$$
where $\delta_{t}^{2}$ is the total solubility parameter and *δ_d_*, *δ_p_*, and *δ_h_* are the cohesive energy densities of the interaction provided by dispersion forces, polar group forces, and hydrogen bonds, respectively. The *δ_t_* have found their greatest use in the evaluation of solvent–polymer chemical interactions and miscibility of different polymers. Recently, *δ_t_* has been employed for elucidating the wettability of drug on polymer surfaces. Thus, this value was experimentally determined for huge numbers of solvents, drug substances, and other chemicals and are collected in the Hansen solubility parameter database of polymeric materials [72,73,74]. The Hansen solubility parameters have been reported to better calculate the drug loading than the F–H interaction parameters [72]. These parameters were also employed to show a relationship between the release rate of drugs from polymer nanoparticles to solubility between drugs and polymer matrix [74,75,76,77,78].

#### 2.3.6. Salting-Out

The salting-out procedure was developed to avoid the use of organic solvents, which are dangerous to the environment as well as to physiological systems. However, recently, this problem was overcome thanks to the introduction of green solvents. The first to introduce a modified version of the emulsion process that involves a salting-out one were Bindschaedler et al. [79]. This technique implies the dissolution of the polymer into the organic solutions which are typically water-miscible [80]. The most common solvents are tetrahydrofuran (THF) and acetonitrile acetone. The aqueous phase includes the salting-out agent (saturated solution of the electrolyte e.g., magnesium chloride and calcium chloride) and a surfactant, such as polyvinylpyrrolidone (PVP) [80]. The oil phase is emulsified in an aqueous phase, under strong shearing force, by an overhead mechanical stirrer. To decrease the ionic strength in the electrolyte, the distilled water is added into the formed O/W emulsion under a magnetic stirrer. At the same time, the hydrophilic organic solvents migrate from the oil phase to the aqueous phase, which results in the formation of the NPs [80,81]. Calcium-alginate nanoparticles (Ca-Alg) crosslinking phosphorylated polyallylamine (PPAA) were prepared through the salting-out technique, and it achieved 82.55% encapsulation of clindamycin drug. The clindamycin-loaded Ca-Alg/PPAA system showed sustained clindamycin release from the carrier [82]. In a recent work, the Zhang group developed poly(trimethylene carbonate) (PTMC) nanoparticles using both the single emulsion method and the salting-out method [83]. They observed that the salting-out method produced PTMC with a size between 183 and 251 nm. Moreover, they demonstrated that in the salting-out method the nanoparticle size was less affected by the stirring speed and polymer concentration than in the single-emulsion method and smaller particles were always obtained under comparable conditions. These dissimilarities are perhaps due to water-miscible nature of the organic solvent used in the salting-out method (THF), while dichloromethane, used in the single-emulsion method, is a hydrophobic solvent. Therefore, the salting-out method requires less energy for making the droplet surface. Allemn et al. developed savoxepine-loaded PLA nanoparticles using a salting-out process by avoiding surfactants and chlorinated solvents. Moreover, they improved the drug loading (9%) and the drug entrapment efficacy (95%) acting on the pH of the aqueous phase by using a savoxepine base rather than the methanesulfonate salt. In vitro release studies have proved that the drug release from the nanoparticles could be modulated from several hours to more than 30 days by changing the mean size of the nanoparticles and their drug loading is able to last, thus allowing the preparation of an injectable extended-release dosage form [84].

#### 2.3.7. Nanoprecipitation

The nanoprecipitation method is a relatively simple and reproducible method. Nanoprecipitation is known as the solvent displacement method or interfacial deposition method [15].

It is a one-step process involving typical solvents, such as acetone, acetonitrile, dimethylacetamide, dimethylformamide, dimethylsulfoxide (DMSO), 2-pyrrolidone, *N*-methyl-2-pyrrolidone (NMP), PEG, and tetrahydrofuran. The oil phase is added to the aqueous phase under mild stirring and the nanoparticles are created in the colloidal suspension. The formation of nanoparticles is fast and easy. There are some critical parameters, such as injection rate of the organic phase, rate of agitation, and ratio between oil phase to the aqueous phase. This method could be applied for hydrophilic and hydrophobic drugs. Polymer and drug are dissolved in a water-miscible organic solvent, for example, acetone or methanol. The solution is then gradually added into an aqueous solution that contains a surfactant. By rapid solvent diffusion, the NPs are generated instantly. Afterwards, the solvents are evaporated under reduced pressure. Recently, Treekoon et al. designed odo-substituted aza-BODIPY (AZB-I) encapsulated nanoparticles via the nanoprecipitation method using the amphiphilic poly(ethylene glycol)-*block*-poly(ε-caprolactone) polymer (PEG-*b*-PCL) for a targeted nano-delivery system that can be triggered by near-infrared (NIR) light. The aza-BODIPY is a near-infrared dye that holds satisfactory properties for biological applications, including minimal auto-fluorescence from biological samples, reduced light scattering, and high tissue penetration. In this work, the nanoparticles (AZB-I@PEG-*b*-PCL) showed a monodisperse spherical morphology with hydrodynamic average sizes ranging from 44.6 to 48.2 nm. Detection of intracellular reactive oxygen species (ROS) in cells as well as the live/dead viability/cytotoxicity assay after near-infrared (NIR) light on 4T1 breast cancer, proved the photodynamic therapy (PDT) efficacy of the AZB-I@PEG-*b*-PCL NPs. These results were confirmed by in vivo PDT in 4T1 tumor-bearing mice; the mice treated with AZB-I@PEG-*b*-PCL NPs at 32 mg kg^−1^ (equivalent to 2 mg kg^−1^ AZB-I) showed 49.8% tumor growth inhibition at day-3 post PDT and tumor growth suppression for up to 14 days post PDT [85]. De Castro et al. developed polymeric nanoparticles (NPs) of poly(lactic-co-glycolic acid) (PLGA) and polyethylene glycol (PEG) loaded with clofazimine (CFZ) for treating central nervous system tuberculosis. The nanoparticles were manufactured by both conventional and microfluidic techniques using the nanoprecipitation principle. CFZ entrapment into PLGA-PEG NPs overcame the poor water solubility and toxicity of the drug that limited its bio-availability. Moreover, to pass the blood brain barrier, the PLGA-PEG NPs were functionalized with a transferrin receptor (TfR)-binding peptide, directing brain drug delivery for treatment by the intravenous route. The TfR-binding peptide-functionalized NPs presented higher cell interaction and upper CFZ permeability across hCMEC/D3 cell monolayers compared to the non-functionalized NP control [86]. Recently, nanoprecipitation has been integrated with microfluidics technologies. Zoqlam et al. compared the nanoparticles prepared by conventional and microfluidics-assisted nanoprecipitation using plasmid DNA-loaded PLGA-Eudragit nanoparticles as the model system. They demonstrated that PLGA-Eudragit nanoparticles prepared by microfluidics were statistically considerably larger than the ones obtained by conventional nanoprecipitation. Moreover, PLGA-Eudragit nanoparticles prepared conventionally presented higher DNA loading efficiency and higher transfection efficiency in respect to the ones obtained by microfluidics [87]. Microfluidics is better explored in the next paragraph while a table (Table 1) summarizing all fabrication techniques and encapsulation methods is attached below.

### 2.4. Microfluidics

It is crucial to produce monodispersed microparticles with uniform size and morphology, in order to have precise control of the release of the drug. To meet these requirements, many efforts have been made to generate droplets with controlled sizes and shapes using traditional methods, such as layer-by-layer assemblies [88], precipitation [89], spray-drying [90], and phase separation. However, these conventional techniques produce drug carriers that have large polydispersity in size, high variability in structure, and a wide range of encapsulation efficiencies. For these reasons, alternative technologies need to be considered to produce highly controlled emulsions. By using a microfluidic platform, it is possible to generate and manipulate droplets that meet the sophisticated requirements in biomedical applications. Microfluidic devices allow the control of fluids at micro and nanometric scales and the precise tuning of the composition and geometrical characteristics of generated microparticles [7,12]. In particular, by regulating the channel geometry and flow rates, microparticles of reproducible and tailored sizes can be obtained [91]. Since monodispersed droplets are usually made by emulsifying two immiscible fluids, the microfluidic chips exploit the flow focusing between a continuous (usually oil) and a dispersed phase (usually water) [7]. When the continuous phase meets the dispersed one, the difference in terms of surface tension leads to the formation of drops. The continuous phase acts as a carrier for the dispersed one, and for that purpose it should contain a surfactant to make the emulsion stable. The size of the microdroplets can be controlled by varying the ratio between the volumetric flow rates of the two phases [91].

Loizou et al. [92] proposed a specific equation to relate the size of the particles to the Reynolds number, to the viscosity of the fluid and to the flow rate ratio between dispersed phase and continuous phase. They used a T junction microfluidic device to experimentally investigate the effect of different flow rate ratios $\frac{Q_{d}}{Q_{c}}$ ($Q_{d}$ flow rate of dispersed phase and $Q_{c}$ flow rate of continuous phase) on the droplet size. By increasing the flow rate of the continuous phase, the volume of the droplet increased.

Another important aspect is related to the rheological properties of the continuous and dispersed phases that can significantly affect droplet formation [93]. Some amphiphilic molecules, such as surfactants, play an important role in preventing droplet coalescence, stabilizing droplet interfaces. These molecules can also alter the interfacial tension between two immiscible phases and consequently change the droplet size and generation frequency. In a dedicated study, different concentrations of Tween 20 surfactant in the dispersed phase were employed and the effect of interfacial tension on W/O droplet formation in a cross-junction microchannel was studied. It was observed that as the surfactant concentration increased from 0 to 1 vol.%, the interfacial tension and droplet size decreased by more than 70% and 120%, respectively [94].

An example of relationship between flow and particles size, together with polymer concentration is summarized in Table 2 below.

Regarding the microfluidic device configuration, droplet-microfluidic devices use specific channel geometries to produce particles. The most common channel geometries are flow focusing, T-junction, and co-flow, whose schematic representations are reported in Figure 2.

#### 2.4.1. Flow Focusing

In flow-focusing geometry (Figure 2a), the continuous phase squeezes the dispersed phase from the top and bottom at a cross-like junction [95]. When the two phases come into contact, shear forces from the continuous phase act on the dispersed phase, allowing the separation of droplets with uniform size. By adjusting the viscosity of the immiscible fluid, the angle at which the phases meet and the flow rate of both continuous phase and dispersed phase, it is possible to modulate the size of the microdroplet and their rate of formation. In flow-focusing design, a wide range of biomaterials can be considered to produce microdroplets, such as PLGA, PEG, PCL, and ethylene oxide, or biological materials, such as liposomes, polymer-DNA, and polymer-lipid nanocomposites. Moreover, organic fibers, such as alginates, chitosan, polyurethane, and poly (methyl methacrylate) (PMMA), have been used as matrices for the fabrication of drug-releasing particles [27].

Kwon et al. presented salient examples of this microfluidic configuration to form drug-releasing polycaprolactone microparticles of sizes between 25 and 160 μm, by means of a flow-focusing junction microfluidic device at a 70 °C [96]. The flow regimes varied from a threading regime at high flow rate ratios to create the smaller sized particles, to a dripping regime where the larger particles were formed. They also developed another protocol in which poly (dimethacrylate-co-trimethoxysilyl propyl methacrylate) hybrid microspheres were fabricated using a microfluidic flow focusing device.

Kim et al. developed a protocol in which poly (dimethacrylate-co-trimethoxysilyl propyl methacrylate) hybrid microspheres were fabricated by the use of a flow focusing device. In this work, highly monodisperse organic microparticles were photo polymerized in situ by combing a UV irradiation module (Figure 3a) [97]. In a work by Here et al., a method was shown for droplet microfluidic synthesis of monodisperse PLGA particles of controllable size with low toxicity solvents and a demonstration of the encapsulation of fluorescein as a drug model and its penetration through animal tissue. They demonstrated the encapsulation of a drug model, fluorescein, and its penetration through the cornea of rabbits (Figure 3a) [98].

#### 2.4.2. T-Junction

The T junction geometry is the simplest and most used geometry for droplet microfluidics; it consists of two channels that intersect to form a 90° angle (Figure 2b). In the orthogonal channel, the dispersed phase flows, whereas the main channel is filled with the continuous phase [99]. When both phases meet at the junction, droplets are formed by means of the shear force applied to the internal phase by the external phase, of their relative viscosities, and of their relative flow rates. Based on the specific coating of the channel walls, different types of emulsion can be produced. O/W or W/O/W emulsions require hydrophilic coating, while hydrophobic channels produce W/O or O/W/O emulsions [7]. In the T-junction configuration, three different regimes of droplet formation can be identified based on the capillary number: dripping, jetting, and squeezing.

Several studies are based on the preparation of microparticles generated by means of the T-junction. One example considers the effect of different viscosities of carrier oil in the W/O emulsion, specifically on how microparticle size and generation rate are affected. Mineral oil with four different viscosities (5, 7, 10, and 15 cSt) and five flow pressure conditions was tested to compare droplet generation under five different flow pressure conditions. Precise size of the droplets decreased when the viscosity of the oil increased regardless of the flow pressure [100] (Figure 3b,c).

Jafarifar et al. prepared risperidone-loaded PLGA microspheres with uniform size, and monotonic and reproducible release profile. Moreover, they compared the PLGA droplet obtained by microfluidics and batch method in terms of encapsulation efficiency and release profile [101]. Another important example in which a T-junction for drug releasing particles is used, is explained by Xue et al. [25]. They worked on the production of 5-Fluorouracil (5-FU)-loaded biocompatible poly(ethylene glycol) diacrylate microspheres that resulted in a monodispersed size distribution, used for sustained drug release. These PEGDA carriers show relatively fast elution in the first 12 h and sustained release over the next 156 h [12].

**Figure 3 pharmaceutics-14-00872-f003:**
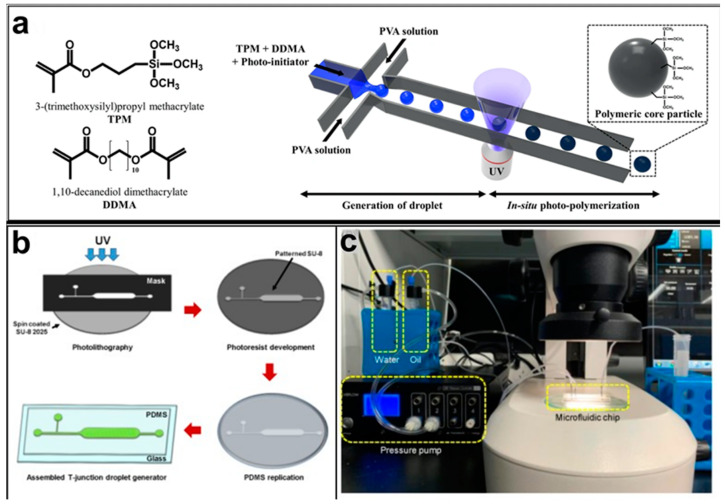
(**a**) Example of a flow-focusing chip to produce photocurable droplets containing the monomers (TPM and DDMA) and the photoinitiator in the continuous phase (PVA solution). Under UV-irradiation, photopolymerization is initiated. Reprinted from reference [97]. Copyright 2018, Scientific Reports. (**b**) The fabrication process of the T-junction chip to generate droplets by varying oil viscosity. (**c**) A photograph of the experimental setup. Reprinted from reference [100], Copyright 2019, Micromachines.

#### 2.4.3. Coaxial-Flow

In a co-flow geometry (Figure 2c), also called coaxial junction, the dispersed phase channel is injected into and aligned with the continuous phase channel and dispersed phase and continuous phase fluids flow in parallel through the channels [102]. Coaxial geometries are characterized by capillaries, mostly made of glass, which can be placed inside one another to generate droplets. When dispersed phase and continuous phase meet, droplets are generated. Droplet formation is obtained by using different flow rates between each of the phases. Indeed, at low flow rates of both the continuous and dispersed fluids, dripping is the predominant regime, resulting in the formation of spherical droplets [103]. At high flow rates, the jetting regime is predominant, and droplets form by breaking a thin stream of dispersed phase farther downstream due to convective instabilities. Typically, this droplet generation device has been used for the production on nano-sized releasing particles.

An example concerns a capillary-based two-phase microfluidic device designed to prepare monodisperse PLGA microspheres to load rapamycin (RAPA). With this platform, RAPA-loaded microspheres have been obtained with an encapsulation efficiency around 98% [104].

Di et al. reported a simple coaxial microfluidic mixing method to achieve continuous production of celecoxib nanoparticles with improved solubility and absorption of drugs in aqueous media. Spherical particles with higher dissolution rate have been produced in a turbulent jet regime [99].

### 2.5. Microneedles

Microneedles (MNs) are sub-millimetric microstructures that can be inserted into the skin with no pain where they can release the encapsulated drug upon degradation or dissolution of polymeric matrix [14,27,105]. In general, the fabrication of microneedles is based on two methods, stamp-based method and stamp-less method. The stamp-based method requires the manufacturing of a master with the desired micropattern from which the stamp can be gained. While the stamp-less method is based on the elastic behavior of materials [106]. From a pharmaceutical point of view, sustained delivery of therapeutic agents is attractive, because such a delivery mode helps to preserve a steady range of drug concentrations in the blood. Considering this, multi-compartmental polymeric microneedles could be very promising to tune the drug release profile. Multi compartmental polymeric microneedles can entrap the drugs into the polymeric particles to provide sustained delivery. On the other hand, a drug may be encapsulated into the polymeric part of the tip to endow burst release of the therapeutic agents [27]. In more detail, a novel bottom-up approach has been proposed for the encapsulation of collagenase protein into PLGA MPs, obtained by a modified double emulsion technique to achieve a tunable porosity, and then loaded into polymeric microneedles for intradermal drug delivery to protect the protein content during storage and their administration [25]. Another recent approach to tune collagenase release consisted of engineering a MN matrix starting from a W/O precursor emulsion formulation containing PLGA and poly(1-vinylpyrrolidone-co-vinyl acetate) (PVP/VA) as polymers and lecithin and maltose as emulsion stabilizers. Microneedles were fabricated by electro-drawing, a stamp-less, fast, mild temperature and one-step strategy. MNinner microstructure analysis, collagenase encapsulation, release and activity of different W/O emulsions were compared to provide an efficient protein release profile [107]. These systems represent different biomedical strategies for treating human diseases with labile therapeutic agents.

Responsive polymeric microneedles release their cargos as a response to an internally or externally generated stimulus. The internal stimuli are physiological signals, such as glucose level of serum, and pH. The external triggers could be UV-light, infrared, and electrical [16].

For instance, insertion responsive microneedles composed of a separable tip and base to increase penetration ability were developed (Figure 4) [108]. Insertion-responsive microneedles are based on adhesion between tip and base, rather than mechanical interlocking. When insertion responsive microneedles are pushed into the skin, mechanical stress is applied, thereby the creation of cracks and separation occurs. Upon removal, the inserted tips are detached from the base and the coated therapeutic agents are released into the skin. Insertion responsive microneedles were designed to provide skin insertion without needing a patch. The impact of base geometry (with or without a walled square pyramid stand) on the mechanical strength of insertion responsive was assessed. Pyramidal and square hyaluronic acid microneedle tips with a polycaprolactone pedestal were fabricated. Upon skin insertion, the tips were detached from the base due to the relatively weak adhesion strength between polycaprolactone pedestal and hyaluronic acid. Ex vivo skin insertion tests proved that following insertion, regardless of the presence of a wall on the stand, separation of the tips from the base happened [105]. However, only insertion-responsive microneedles with the single walled square pyramid stand were thoroughly implanted into the skin. Mechanical testing results exhibited that the presence of a wall enhance the mechanical stability of the microneedles. The presence of the wall also allowed for suitable adhesion between the tips and pedestal, avoiding tip breakage during indentation, while still allowing the tip to be separated from the baseplate upon removal.

To ameliorate the performance and enhance the mechanical adhesion of microneedles, the next generation of microneedles has emerged [109]. The outstanding features in natural species offer promising characteristics to design bioinspired or biomimetic microneedles. For instance, the structure of the teeth of limpets [110], eagle claws [111], grooved fangs of snakes [17], and ice [112] are a great source to design microneedles with greater properties.

Due to the elasticity of the skin, polymeric microneedles must be strong enough to overcome this feature to have a successful penetration. In light of this, ice-inspired microneedle patches were developed (Figure 5) [111]. Frozen microneedles inspired by naturally occurring ice were able to completely penetrate into the skin. To compare the mechanical strength of the microneedles with or without the freezing process, different types of materials including water, Matrigel, GelMa, and alginate were assessed [112]. It was demonstrated that unfrozen GelMa and alginate microneedles were not able to be thoroughly extracted from the mold and water and Matrigel could not even create the shape of needles. While all frozen material could completely be extracted from the mold with an intact microneedle shape. The implantation abilities of different microneedles were subsequently assessed. Results proved that unfrozen microneedles barely perforated the skin models. On the contrary, frozen microneedles could entirely puncture both agarose and porcine skin [112].

Inspired by the grooved fangs of rear-fanged snakes with a rapid capillary force–driven delivery system, a bio-inspired patch including open grooves on the surface of needles was developed [17]. Such a bio-inspired platform can deliver various therapeutics in liquid form, applying mild thumb pressure without needing a complex pumping system. Tri-, tetra-, penta-, and hexa-open grooved microneedle patches were fabricated using the photolithography technique. The bioinspired microneedles that contained more grooves were taller than those with fewer grooves. Additionally, all needles were capable of puncturing mouse skin. The maximum depth of needle insertion was obtained by the hexa-grooved BMN patch. Since deeper penetration depth could result in pain, the in vivo study was carried out on the penta-grooved microneedles. Penta-grooved microneedles possess sufficient penetration depth and channel size for efficient drug delivery. Applying grooved fang-inspired microneedles to deliver ovalbumin and influenza induces an immune response in pig and mice.

Although microneedle patches play a crucial role, particularly in transcutaneous drug delivery, their frequent usage may cause skin irritation to a certain extent. A possible solution can be represented by nanoneedles which have the same working principle as MNs, but a smaller size, around nanometer-size (sub-100-nm diameter) that help to pierce the cell membrane and transfer biological cargos directly into the cell interior, reducing skin irritation [113].

### 2.6. Polymer Fibers by Electrospinning

As well as micro/nanoneedles and micro/nanoparticles, micro/nano fibers, thanks to their porous structure, can incorporate several drugs, such as antibiotics, anticancer drugs, proteins, and also nucleic acids and therefore can be largely employed for drug delivery in therapy [114], regenerative medicine [115], and transdermal drug delivery [116].

Electrospinning is probably the most employed fabrication method for producing fibers having diameters in the order of microns or nanometers [20]. While conventional fabrication methods, for example, mold-based synthesis or melt blowing, present a considerable difficulty in the production of micro/nanofibers since they lack homogeneity in their structure and in the distribution of the drug, electrospinning allows the manufacture of fibers starting from polymeric materials in a continuous, simple, and cost-effective way [18,114].

The typical components of an electrospinning system are shown in Figure 6a. They include a flow regulation system, which is generally represented by a volumetric pump; a syringe containing the polymeric material that will be transformed into micro/nanofibers; a collector on which fibers are deposited, and a high voltage generator. During electrospinning, the polymeric material, brought to a fluid state by melting or dissolution in suitable solvents [117,118], undergoes electrification by application of high voltages: when the electrostatic repulsion of the charged polymeric solution exceeds the surface tension, a Taylor’s cone is formed and the jet is generated [19]. This jet therefore undergoes acceleration and elongation through the application of an external electric field. As it moves towards the collector, the jet thins, solidifies, and settles on the substrate placed on the collector [18] (Figure 6a–c).

The parameters that play a crucial role in the structural characteristics of the fabricated fibers can be classified in (i) environmental parameters, such as temperature, which has a great influence on the viscosity of the solution, and on the evaporation rate of the employed solvent. Another relevant environmental parameter is represented by the humidity, which mainly affects the solidification process of the solution exiting from the syringe [123]; (ii) process parameters, such as the generated electric field, which significantly affect the diameter of the produced fibers, or the flow rate that determines changes in morphology and uniformity of the fibers [124]. Finally, the orientation of the support placed on the collector can greatly influence the characteristics of the nanofibers deposited on it [122]; and (iii) solution parameters, such as chemical characteristics of the polymers and of the solvents employed to produce the nanofibers, final viscosity, conductivity, and surface tension [121].

Solutions prepared for electrospinning can include two or more polymeric materials or other additives as particles or drugs. Based on the solution characteristics the most appropriate electrospinning method can be chosen (Figure 6d). In blend electrospinning the bioactive components are dissolved or degraded in the solution [114,125]: drugs and one or more polymers are mixed together in the same solvent system, with a very high drug loading capability [126,127]. This methodology has been widely employed to create nanofibers for the controlled release of drugs, such as 1,3-bis (2-chloroethyl) -1-nitrosourea from implantable biodegradable systems [128], doxorubicin for the treatment of liver cancer [129], or paclitaxel loaded on poly-(d,l-lactide-co-glycolide) fibers to treat malignant brain tumors [130].

Coaxial electrospinning consists of using a coaxial needle and two distinct solutions to realize nanofibers typically employed to encapsulate and protect the drugs in the core through the outer shell of the polymer [121]. This method represents a possible solution to prevent the initial burst release [20], as demonstrated in [83] which the authors found the best way to load fluorescein isothio-cyanate-conjugated bovine serum albumin in a water-soluble core of poly(ethylene glycol) within a poly(ε-caprolactone) (PCL) shell. Emulsion electrospinning uses the same set-up of the blend electrospinning, but the solution is composed of a continuous phase, subject to rapid evaporation during the electrospinning, and the aqueous phase containing the active components. After the evaporation of the continuous phase, the aqueous phase droplets are aligned and directed towards the center of the jet by the electric field [131].

The choice of the loading method depends mainly on the characteristics of the drug, specifically on its solubility in the polymeric solution employed for the production of the nanofibers. In general, hydrophobic drugs, such as doxorubicin and paclitaxel, are solubilized in organic solvents [132]. Hydrophilic drugs, such as ampicillin, peptides, and proteins, are water-soluble and are better processed in water-soluble polymers [133]. On the other hand, the method of loading the drug is decisive in the release process [20], as well as the interaction between nanofibers and solvent [134].

The characteristics of the deposited nanofibers are also influenced by the position of the collector (Figure 6e), which can be placed vertically, horizontally, or opposite to the tip of the syringe. In vertical electrospinning, which represents the most frequent set-up, the tip of the syringe is positioned vertically to the collector, and the force of gravity acts on the jet exiting the syringe, speeding up the elongation and deposition of the material [135]. Conversely, in horizontal electrospinning the tip of the syringe and the collector are positioned horizontally on the support surface. This set-up requires applications of higher electric fields to overcome the gravitational force that tends to deflect the jet downwards [135]. Finally, another configuration exists, converse electrospinning, in which the tip of the syringe and the collector are positioned in opposite position with respect to vertical electrospinning. Here, the electric fields used are higher than in the first two techniques and the obtained nanofibers are thicker and distributed in a narrower region [136].

Recently, a specific electrospinning method was developed in [137] to produce polycaprolactone (PLC) nanofibers loaded with magnetic nanoparticles, used as thermal mediators selectively activated under an alternating magnetic field. The aim of the work was to combine the chemotherapy with magnetic hyperthermia to maximize the effect of the treatment. Specifically, magnetic iron oxide nanocubes (IONCs) and doxorubicin (DOXO) hydrochloride were added to a PCL solution in chloroform obtaining drug-loaded magnetic fibers (Figure 7a). The magnetic properties of these new nanofibers showed an increase in the temperature profile in the presence of a safe magnetic field, which in combination with the drug, ensured an improvement of the cytotoxic effects.

Kang and co-workers [138], implemented an energy-saving electrospinning process for the efficient fabrication of the PVP nanofibrous films loaded with ketoprofen. The newly proposed electrospinning device is composed of the power supply, two-fluid drivers, a fiber collector, and two homemade spinnerets, denoted as N1 and N2. N1 is a classical spinneret, while N2 denotes a concentric spinneret with a Teflon-core rod designed to conduct the energy-saving process (Figure 7b). The nanofibers thus produced showed characteristics similar to those generated with the classic methodology but were generated consuming only 53.9% of the electric energy.

However, apart from the significant benefits experienced from fibrous matrices used for drug delivery, as high surface to volume ratio providing great drug release, they present an initial burst release of the drug, due to a rapid drug diffusion; frequently, the low interactions between drug and polymer determine a prevalent localization of the drug near the surface of the nanofibers, and consequently, its rapid diffusion into the surrounding aqueous solution [20].

A possible solution is represented by the introduction of micro- and nanocarriers into the fibers, e.g., Zheng and co-workers proposed a blend of PLGA and amoxicillin (AMX)-loaded hydroxyapatite (HA) nanocarriers to produce morphologically more uniform and smooth nanofibers, with a much longer release profile and that are biologically cyto-compatible [139].

Another crucial point consists of the effect on drugs of some particularly aggressive solvents, employed for the dissolution of polymers. In this regard, there are several techniques of immobilization of the drug on the surface of already electrospun fibers that avoid its denaturation and inactivation: (i) physical methods, which exploit the different interactions that can be established between drug and fiber surface, such as electrostatic, hydrophobic, or Van der Waals interactions [140], (ii) chemical methods, which have a better ability to control the amount of the incorporated drug, and are based on the addition of particular functional groups, such as amine groups, as reported in [141] where methotrexate (MTX) has been immobilized on the surface of iron oxide nanoparticles via a self-assembled poly(ethylene glycol) monolayer (PEG SAM).

Finally, several studies are focused on the development of new techniques aimed at improving the quality and functionality of the fibers produced by electrospinning: several research lines study how to generate smart fibers, capable of releasing the drug in response to different stimuli [142,143].

## 3. In Silico Models

Mathematical modeling applied to drug delivery and its release represents a new and promising scientific field, from which medicine [144,145], the pharmaceutical industry, and research [146], can greatly benefit. Taking advantage of the vast knowledge on well-known physical processes, such as diffusion, degradation, a model appropriately designed and correctly defined from a mathematical point of view can, in short time and with relatively low costs, bring improvements in the design of new [143] therapeutic plans that optimize the administered dose, improving the results of the treatment. This can be easily obtained by exploiting the knowledge on the type of drug used and its release profile, as well as all the information and characteristics of the device employed for its delivery [22,23]. The process that leads to the design of a mathematical model passes through the following steps [147]: (i) assumption and selection of appropriate control variables, such as the rates of polymer degradation and drug diffusion [148]; (ii) derivation of a system of equations describing the observed process, specifying appropriate state variables and parameters. The granularity of a mathematical model varies according to the available knowledge about the process or analyzed system, as well as to the assumptions that lead to the definition of the model itself. Generally, this granularity is strictly related to the accuracy and to the complexity of the devised model [147]; (iii) validation of the proposed model, as well as assessment of the obtained results through the fitting of the experimental data.

Most of the release processes from polymer-based materials can be described by diffusion phenomena, which are mathematically represented by the two laws proposed by Fick [149],
(3)$$J=-D\frac{\partial C}{\partial \chi },$$
(4)$$\frac{\partial C}{\partial }=-D\frac{\partial^{2}C}{\partial^{2}\chi^{2}},$$
where *J* represents the flux per unit area, while *C* is the drug concentration affected by the diffusive processes. These two laws allow identifying of a first important parameter common to most models describing this process: the diffusion coefficient, *D*, depending on typical characteristics of material or drug, temperature, and pressure. All the other model parameters will depend on (i) the theory chosen to define the model, and (ii) the geometry and materials of the device exploited for the drug delivery.

Especially, it is possible to identify the following classes of parameters, based on:Geometry: such as, length, height, radium, and the thickness of the membrane. The main geometric parameters are shown in Figure 8;


Material exploited to realize the devices for drug delivery: such as polymer molecular weight, viscosity, degree of crystallinity, parameters related to the interactions between polymer and solvent (e.g., surface erosion rate constant [151], and temperature), or to the polymer swelling and dissolution;Drug: solubility and diffusion constant size of the drug particles [152], degree of affinity between drug and medium [23].

Several examples of mathematical models applied to drug-delivery are available in the literature: Rothstein et al., exploited mathematical modeling to design new PLGA microparticles starting from the choice of the appropriate material up to the definition of the best conditions for their design and production [153]. In this specific case, the authors analyzed the delayed-release profile able to induce delivery of ovalbumin (OVA) or ovalbumin with the adjuvant alum (OVA-alum) over 1, 2, or 4 weeks from the hydration. The geometry of the microparticles was suitably designed in order to avoid phagocytosis and inflammatory problems and to minimize the initial bursting of the antigen.

Sebe et al. [154] analyzed, through mathematical modeling, the possibility of designing appropriate PVA nanofibrous sheets capable of maintaining a constant local concentration of colistin sulfate. Every single fibrous layer was created by electrospinning, and the final scaffold consisted of a disc of 35 mm diameter. Several solutions obtained by alternating drug-loaded layers (CEL) and membrane layers (MEL) were tested, and the CELMELCELMELCELMEL solution showed an almost constant gradual release over time.

Similarly, in [155] the authors characterized the release profile of artemisinin (ART), loaded on electrospun nanofibers, with the aim of realizing appropriate transdermal drug delivery systems (TDDS) to deliver drugs through the skin of patients with malaria. Several ART-loaded PCL/Col characterized by different polymer ratios were tested to identify the correct strategy to design nanofibers for a controlled release.

## 4. Different Approaches for Mathematical Modelling

It is possible to distinguish between two great classes of mathematical models, based on the following two main theories: mechanistic realistic and empirical/semiempirical theories.

### 4.1. Mechanistic Realistic Theories

Mechanistic realistic theories are based on equations describing real phenomena, such as diffusion, degradation, dilution, swelling, precipitation, or erosion [147]. Models based on these theories are accurate, reliable and their predictive power is very high. Since very often these models are composed of differential or partial differential equations, their computation requires initial and boundary conditions, such as initial drug distribution on the device, or size of the initial surface exposed to erosive phenomena. On the other hand, in silico simulations can be computationally complex and heavy. Some examples of mechanistic models are based on:Diffusion: the drug release is controlled by the diffusion processes (Figure 9a). For this type of model, the initial conditions are represented by the concentration of the drug at the beginning of the release processes, while the boundary conditions are assumed for surface, mass, and volume of the surrounding medium. It is possible to distinguish between reservoir, monolithic, and miscellaneous systems, based on the strategy employed to deliver the drug, each of which with specific initial and boundary conditions [22,147,151,156].Swelling: this mechanism allows better control of the release of a specific drug. There are two important consequences of the polymer swelling, which concern the increment of the diffusive path due to the decrease of the concentration gradient, and, increment of the mobility of the macromolecules with an improvement of the drug release (Figure 9b) [147,157].Degradation/erosion: where the term degradation refers to the formation of oligonucleotides or monomers by polymer cleavage, while the erosion refers to the process of material loss (Figure 9b) [147,151,158].

Casalini and co-workers [160] presented a mechanistic mathematical model describing the diffusion processes following the degradation of polylactic-co-glycolic acid (PLGA) microparticles. The choice of this type of microparticle was justified by their excellent biocompatibility and biodegradability, as well as by the possibility of an easy control of the drug release rate. The authors designed and validated a mathematical model able to describe the degradation and release of a particular drug contained in the PLGA microparticles. The degradation of the microparticles was modeled through mass conservation equations describing PLGA hydrolysis kinetics and autocatalytic behavior, while drug release was described through dissolution and diffusion of dissolved active principle through the polymeric matrix. Fickian behavior was chosen to describe the diffusion process of oligomers, water, and drug. Every single parameter was associated with a specific physical meaning, such as polymer degradation or transport phenomena, and it was estimated from the experimental dataset of three different drugs: paclitaxel, lidocaine, and ibuprofen. Mathematical modeling allowed the authors to investigate the implication of geometric parameters, such as radium, in the degradation and diffusion of the drugs.

### 4.2. Empirical/Semiempirical Theories

Differently from mechanistic realistic theories, empirical/semiempirical models are not based on a complete mathematical definition of the real process, but, rather, on a descriptive representation of the observed phenomenon [22,23]. The level of approximation introduced by these theories makes the derived models unsuitable for prediction. However, unlike the mechanistic models, the empirical/semi-empirical ones are easier and computationally less heavy.

Some of the most important kinetics are reported below [23]: (i) Zero-order kinetic release: the dissolution of the polymer is the only factor that appears to control the drug release rate [23,161,162,163]. Zero-order kinetic release shows a drug release rate constant over time. A simple zero-order release kinetic can be represented by the following model
(5)$$M_{t}=M_{0}+k_{0}t,$$
where $M_{t}$ represents the amount of drug released or dissolved, $M_{0}$ is the initial amount of drug in solution, and $k_{0}$ is the zero-order release constant. A typical description of the drug release profile, obtained through this kinetics, is shown in the Figure 10a.

(ii) First-order kinetics release: the release of a drug is driven by the difference between its concentration inside and outside the capsule, and it can be represented by the following equation
(6)$$M_{t}=M_{0}e^{-k_{1}t},$$
where $k_{1}$ is the first-order rate constant. In this kinetics, the reaction rate appears to be proportional to the concentration of only one chemical reagent (Figure 10b) [23,163].

(iii) Higuchi model: it is the most famous equation employed to model the release kinetics of a material placed in an infinite solution [164]. It is based on some assumptions, such as one-dimensional diffusion, initial concentration of the drug higher than its solubility, no dissolution, and swelling.
(7)$$M_{t}=K_{H}\sqrt{t},$$
where $K_{H}$ represents the dissolution constant (Figure 10c).

(iv) Korsmeyer–Peppas model: or power law model, it is used when the release mechanism is not completely known [23],
(8)$$\frac{M_{t}}{M_{\infty }}=kt^{n},$$
where $\frac{M_{t}}{M_{\infty }}$ represents the release fraction at time *t*, while *k* is the rate constant and *n* is the release exponent (Figure 10d).

(v) Weibull model: this empirical model was exploited to model a fraction of drug release as a function of time. Having no solid kinetic basis, this model presents several pitfalls that limit its application to the comparison of the release profiles of matrix systems. An example of Weibull model is shown below,
(9)$$M_{t}=M_{\infty }\left(1-e^{\frac{{\left(t-t_{0}\right)}^{\beta }}{\tau_{d}}}\right),$$
where $M_{t}$ and $M_{\infty }$ represent the dissolution at t and at infinite time, respectively, with $t_{0}$ the lag-time of dissolution, $\tau_{d}$ the time corresponding to the 63% of dissolution, and β the shape of the curve [154].

Versatility, ease to treat, and relatively low costs make in silico modeling one of the main promises in pharmacokinetics. The availability of well-posed and complete mathematical models capable of correctly describing the release process represents a powerful tool for the design and development of a drug. This will improve the specificity and efficacy of the drug, allowing for the planning and choice of the appropriate dosage for each single patient [165]. However, several aspects still need to be improved, thus obtaining models able to give information on the spatial distribution of a specific drug at the level of the organs and tissues other than those of interest. This could help in understanding and analyzing side effects.

On the other hand, more complex models would need new computational methodologies that are robust and suitable to obtain reliable results in less time and with less expenditure of resources [166]. Another crucial aspect for the definition of a successful model concerns the parametric identification part—numerous studies, including [167], have highlighted the need to identify parameters with physical meaning and independence from geometry.

## 5. In-Silico Methods to Assess the Performance and Design of Polymer-Based Drug Delivery Systems

In silico methods found applications not only in modeling the dynamics of drug release, but they represent relevant strategies in the analysis and study of what happens at the interface between NPs and the external environment [168]. In this regard, several studies have shown that nanoparticle-based nanomedicines, designed ad hoc to fight diseases, such as cancer, fail during clinical trials; one of the main reasons consists of the lack of knowledge on how these nanoparticles interact with the biological environment, i.e., how they overcome the transport barriers in the body, their circulation in the bloodstream, extravasation in tumors, internalization in target cells, and release of their active load [169]. Through the definition of NP properties related to the interactions with the biological environment, it is possible to optimize their design and production, improving their surface functionalization, and the aspects related to the control and prediction of relevant biological processes, such as the endocytosis. In this case, a strong correlation with the size of the nanoparticles was observed and several mathematical models were proposed. Gao et al. [170] developed a model to describe this dependence; the model shows how, for a specific size, the time of cell internalization assumes a minimum value that diverges (i.e., no endocytosis) for all the nanoparticles with a size outside a specific interval. Furthermore, the shape of the nanoparticles can affect endocytosis [171]; they have been demonstrated as non-spherical nanoparticles, having several meta-stable states, showing a significantly slowed absorption process compared to rod-shaped nanoparticles, which show a much easily internalization process. In addition, a recent study investigated the relationship between size and surface charge of the nanoparticles [172]; the authors proposed a new framework based on a stochastic geometric model of brain white matter (WM), and a mathematical particle tracking model, to tune the efficiency of the transport and diffusion of nanoparticles through brain-blood barrier, acting on size and surface charge. The authors found a positive correlation between diffusion, size, and charge for negatively charged NPs.

The solubility of the drug is another important aspect that can be easily mathematically modeled [173,174]. This parameter allows describing of drug-nanoparticle iterations, and therefore the effectiveness of the treatment that exploits the nanoparticle itself. The classical models, based on the solubility parameters of Hildebrand and Hansen [175], and of Flory-Huggins [176], have the advantage of easily describing the solubility of a drug but they show various limitations that make them inadequate for the prediction of polymer-drug interactions; indeed, these models do not consider all the components of free energy.

Aqueous solubility of a drug can be influenced by several processes, such as absorption, distribution, and clearance, with implication on the real efficacy of the drug. In silico approaches represent useful tools to predict the aqueous solubility, and can be based on (i) experimental measurements [177], (ii) 3D parameters depending on molecular stereochemistry [177], and (iii) fragmental and atom-type methods using 1D–2D parameter [178].

Furthermore, dialysis plays a crucial role in the dynamics of drug release. Dialysis-based methods are used to determine the drug release profiles and kinetics, in order to guide the development of NPs [179]. However, processes, such as centrifugation, could influence dialysis, since external centrifugal forces could alter drug release dynamics. Several mathematical models have been proposed to analyze the mass transport during the dialysis, assuming a homogeneous distribution of the drug among the compartments under examination [180].

In order to optimize the production of NPs while improving their functionality, quality by design (QbD) approaches have been shown to be particularly useful, since they define standardized procedures for the optimized design [181] and production of NPs, with considerable improvements in the therapeutic outcome [182]. In this regard, Yeom et al., proposed a combinatorial approach which consists of QbD and discrete element method (DEM) to develop an adequate scale-up strategy for the blending process of an amlodipine-based solution. Another example is given by Shang et al. [183], which presented the pharmacokinetic (PK) of carbamazepine (CBZ) through ACAT model, a particular type of PK-model exploited to describe the kinetics of the oral drug from the administration to the dissolution, implemented in QbD, with the aim of understanding the drug formulation, and its release mechanisms.

Mathematical modeling represents an important tool thanks to it being possible to extract important information, difficult to observe otherwise, which can be used to understand the dynamics of drug release, as well as to optimize the production of NPs and improve their functionality.

## 6. Polymer-Based Micro and Nano Matrices for Drug Delivery on Market

The translation of bioactive molecules into commercial drugs requires the fulfillment of several requirements. In the treatment of diseases and injuries, therapeutic molecules, regardless of their composition, chemical or physical nature, must satisfy parameters of biocompatibility, stability under physiological conditions, specificity for the target, and minimal adverse effects.

An interesting strategy to overcome these problems involves the controlled release of drug-polymer conjugates, through changes in pH, temperature, enzymatic concentration, or attack of specific ligands.

Based on their mechanism of action, polymer-based drug delivery systems used in the pharmaceutical market could be divided into two principal groups: (i) reservoir-based systems and (ii) polymer-drug conjugated systems [184].

The first group concerns reservoir-based systems. They could be external to the body or implanted, and are particularly used for long-term administration of a drug localized to a specific area that is either difficult to reach via systemic administration (e.g., eye, body cavity, etc.), as drug depot for long-term systemic administration, or in combination.

In polymer-drug conjugated systems, several copies of bioactive agents (e.g., small molecules, oligosaccharides, peptides, etc.) are chemically linked to a polymer scaffold. Based on the properties of the polymer carrier, polymer–drug conjugates can improve drug solubility, half-life, and drug targeting.

Based on their formulation they could be classified into implants, microneedles, nanoparticles, etc. [185].

Currently, polymer-based drug delivery systems are increasingly used in several biomedical fields. Here we focused our attention on some relevant polymer-based drug delivery systems that already reached the pharmaceutical market (Table 3), highlighting their chemical characteristics and their biological advantages.

### 6.1. Polymeric Nanoparticles

The first FDA-approved controlled-release system was Lupron Depot, a gonadotropin releasing hormone (GnRH) agonist used for the treatment of prostate cancer. It is constituted of injectable microspheres composed of PLA-PLGA copolymer and leuprolide acetate. PLA is characterized by long degradation rates, and thanks to this property, it has been employed in the preparation of the 3, 4, and 6-month formulations of Lupron Depot [186].

Another marketed product based on PLA microparticles is Sculptra, a cosmetic product for the treatment of facial lipoatrophy in people living with HIV, approved in 2004 by the FDA. In 2009, it was FDA-approved under the brand name Sculptra Aesthetic for the treatment of deep facial wrinkles and folds for people with healthy immune systems. Chemically, Sculptra is an injectable cosmetic filler that contains microparticles of PLA, a biocompatible synthetic substance that stimulates collagen production. After injection, the hydrophilic material is slowly resorbed and cleared by macrophages. During the resorption process the PLA microparticles are able to stimulate collagen production, which leads to augmentation of the treated site [187].

Zilretta is another polymeric-based drug approved in October 2017 for the treatment of osteoarthritis knee pain. It is a suspension of microspheres characterized by the presence of a PLGA matrix capable of trapping small crystals of triamcinolone acetonide inside it. Prolonged release of the formulation is ensured by nanochannels on the surface of the microsphere that limit the release of triamcinolone acetonide [188].

The use of polymers is a strategy also used to create new controlled-release formulations of drugs already on the market. An example is Bydureon, a prolonged-release formulation of exenatide. Exenatide is a GLP-1 receptor agonist that shows activity similar to glucoregulatory, such as incretin hormone, but is not significantly degraded by dipeptidyl peptidase-4 (DPP-4), which efficiently degrades native GLP-1 in vivo. It was approved by the FDA in 2005 under the brand name Byetta as an immediate-release aqueous formulation. To increase the duration of action and reduce doses, in 2011 the European Commission approved Bydureon BCise (exenatide sustained release). In October 2017, Bydureon BCise received approval in the US for use in adults with type 2 diabetes. Bydureon formulation uses biodegradable PLGA microspheres which entrap exenatide and provide prolonged release of the peptide over days. It is supplied as a vial and prefilled syringe ‘single dose tray’ or as a prefilled pen ‘dual-chamber pen’. Once injected subcutaneously, the polymer biodegrades over time, thereby releasing the bioactive peptide for absorption into the systemic circulation.

Another widely used strategy to obtain polymeric nanoparticles involves the use of PEG (in pharmacy called macrogol) polymer chains.

An example of this is the FDA-approved drug Adagen. Chemically, Adagen is a PEGylated adenosine deaminase (PEG-ADA, pegademase) nanoparticle approved for the treatment of severe immunodeficiency disease caused by ADA deficiency.

In line with this approach, another FDA-approved PEGylated nanoparticle is Cimzia (certolizumab pegol). It was approved in 2008 and is used to neutralize tumor necrosis factor-alpha (TNF-α) activity. Cimzia is a PEGylated Fab fragment that specifically recognizes and binds to TNF-α, inhibiting its functionality. It is currently used to treat arthritis in rheumatoid, psoriatic, and ankylosing spondylitis.

Neulasta, a PEGylated form of filgrastim, was approved by the FDA in 2002 for chemotherapy-induced neutropenia. Neutropenia (low white blood cell counts) is a common adverse effect found in patients with non-myeloid cancer who receive chemotherapy [28,189].

Another already approved PEGylated nanoparticle is Plegridy. It contains the active material PEG-IFN-β-1a, a glycosylated recombinant PEG-conjugated IFN-β modified with a single, linear 20 kDa methoxy-PEG-O-2-methylpropionaldehyde molecule (mPEG). In 2014, It was approved by the FDA to treat recurrent remitting multiple sclerosis in adult patients [28,190].

### 6.2. In Situ Gelling Systems

Another class present on the pharmaceutical market regards polymer-based injectable implants. Eligard and the Zoladex are two polymer matrix-based sustained release formulations FDA-approved.

Eligard is a medicine for managing advanced prostate cancer. It is an in situ forming injectable depot implant for the delivery of leuprolide acetate, and is packaged as two separate syringes whose contents are combined directly prior to SC injection. One syringe contains leuprolide acetate and the other contains the Atrigel delivery system [191], a matrix composed of PLGA dissolved in a biocompatible solvent system as N-methyl pyrrolidone (NMP). The two components are combined immediately before injection and form a solid implant at the injection site [192].

The Zoladex implant is used in men to treat prostate cancer or in women to treat certain breast cancers or a certain uterus disorder (e.g., endometriosis). It is supplied as a prefilled syringe that contains goserelin acetate dispersed in a cylindrical PLGA matrix. Generally, it is administered by SC injection in the abdomen [193].

Another gel depot delivery system is OncoGel. It is composed of paclitaxel and a PLGA-PEG thermosensitive polymer system (named ReGel) and is used for injection in tumors and release of paclitaxel over a period of 6 weeks. OncoGel injection physically targets paclitaxel to the tumor site while limiting its circulation, resulting in an acceptable safety profile [194,195].

### 6.3. Implants

Compared to other reservoir-based drug delivery systems, implants are less common. However, a quite important application of these drug release implants has been in the ophthalmology field. This is due to the fact that the intraocular structures are easily accessible and are confined and isolated from the circulation by the inner and outer blood-retinal barriers. Currently, there are three already FDA-approved implants for the eye: Retisert, Vitrasert, and Ozurdex.

Retisert, is an intraocular implant used for the treatment of noninfectious uveitis. The implant is constituted by fluocinolone acetonide in a tablet reservoir. This reservoir system contains microcrystalline cellulose, magnesium stearate, and PVA. The tablet is coated on a silicone elastomer cup containing a release orifice and a PVA membrane placed between the tablet and the orifice. Retisert is implanted into the eye and releases fluocinolone acetonide into the vitreous of the eye. The release is controlled by the PVA membrane and decreased over the first month to a steady state [196,197,198].

The second FDA-approved implant is Vitrasert. It is another controlled-release intraocular implant that contains ganciclovir dispersed in a PVA/ethylene vinyl acetate (EVA) matrix. Vitrasert is used to treat cytomegalovirus retinitis, one of the most common ocular infections in AIDS patients. The ganciclovir tablet reservoir is covered by PVA and EVA polymers and attached to a tablet inserted into the eye. The presence of PVA guarantees prolonged drug release while the EVA controls the surface area of the device where ganciclovir can penetrate. The drug release into the vitreous of the eye can last for approximately 6–7 months [199,200].

The third implant is Ozurdex. It is a sterile, biodegradable intravitreal sustained-release implant of dexamethasone that provides sustained delivery of dexamethasone to the retina and vitreous for the treatment of macular edema and noninfectious uveitis [201,202].

### 6.4. Polymer Microneedles

The V-Go disposable insulin delivery device from Valeritas (a BioValve subsidiary) utilizes tapered-cylinder, hollow microneedles and was licensed from Georgia Tech (Reservoir-Based Drug Delivery Systems Utilizing Microtechnology). The V-Go Wearable Insulin Delivery device features an injection molded cartridge made of TOPAS cyclic olefin copolymer (COC) from TOPAS Advanced Polymers. TOPAS COC resins are a chemical relative of polyethylene and other polyolefin plastics.

The V-Go has been available for several years and is used primarily by patients with type 2 diabetes. The patient can use the device for only a single day. The overall size is inches, and it weighs less than 2 ounces, filled with insulin. It comes in three models, each with a different basal rate: 20, 30, or 40 units/d. To deliver a bolus, the patient presses a lock release, then a delivery button, delivering two units. The lock release must be pressed before each 2-unit delivery. The device is capable of delivering a 2-unit dose 18 times per day. Thus, the maximum daily dose is 76 U (basal of 40, bolus maximum of 36). It is approved for both U100 lispro and U100 aspart insulins (Patch Pumps for Insulin 2018).

Another implant approved by the FDA is the Sanofi’s Fluzone Intradermal Quadrivalent [203]. It has been approved for adults from age 18 to 64; it is administered directly into the skin of the arm through a microneedle that is one-tenth the size of the typical needles used for intramuscular injection of flu vaccines. Fluzone Intradermal Quadrivalent contains four killed flu virus strains and it is indicated for active immunization for the prevention of influenza disease caused by influenza A subtype viruses and type B viruses contained in the vaccine [203].

### 6.5. Electrospinning Based Nanofibers

Another class present on the pharmaceutical market regards electrospinning-based nanofibers. The first example of an already FDA-approved compound is Rivelin. It is a muco-adhesive two layered patch comprised of a muco-adhesive, drug-delivery layer, and a protective layer that delivers a pharmaceutical product directly to wet tissue surfaces. Rivelin utilizes a unique patch technology that adheres to mucosal surfaces for extended periods, facilitating selective delivery of a pharmaceutical agent to the target site of action impacting disease progression, while limiting delivery to surrounding areas. Rivelin Clobetasol patch (“Rivelin-CLO”), is the first biodegradable oral adhesive patch designed for local delivery of clobetasol to treat symptomatic oral lichen planus (OLP) lesions. It is currently under Phase II clinical trials. The therapeutic patch utilizes electrospinning technology, and it can adhere to the oral cavity for an average of 90 min, and other wet tissue surfaces for approximately 9 h, while delivering a steady therapeutic dose to the lesion [204]. Another already FDA-approved medicament is SurgiClot. It is the first and the only fibrin sealant patch designed specifically for bone bleeding. It consists of two components: non-woven nanofiber dextran and two clotting proteins, thrombin, and fibrinogen. The dressing is constituted of five layers of dextran that encapsulates lyophilized both human proteins. Electrospun dextran fibers can deliver a bolus of proteins to promote blood clotting. In detail, upon topical placement, the nanofiber dextran adheres to the surface to form a physical barrier and once in contact with blood, dextran starts to dissolve releasing the two proteins that form an insoluble fibrin clot (SurgiCLOT-St Reresa Medical. Available online: https://stteresamedical.com/technology/surgiclot, accessed on 23 December 2021) [205].

## 7. Conclusions

Drug efficacy usually depends on the ability to be released in a spatially/temporally controlled way and for this reason, different drug carriers are being developed. Among these, polymeric carriers are becoming increasingly important in biomedicine thanks to their ability to be biocompatible and biodegradable. To produce these nano/micro systems, various manufacturing processes are currently employed, comprising an array of different techniques as single and double emulsion, layer by layer, precipitation, microfluidics, electrospinning, and so on. The selection of the appropriate technique enables the manufacture of a broad range of particles with different morphologies, porosities, or size distributions.

On the other hand, the development of optimized carrier materials for the encapsulation of drugs with high efficiency is frequently hindered by the need for trial-and-error experiments. In this context, combining experimental and in silico approaches can be a promising way to overcome this shortcoming.

## Figures and Tables

**Figure 1 pharmaceutics-14-00872-f001:**
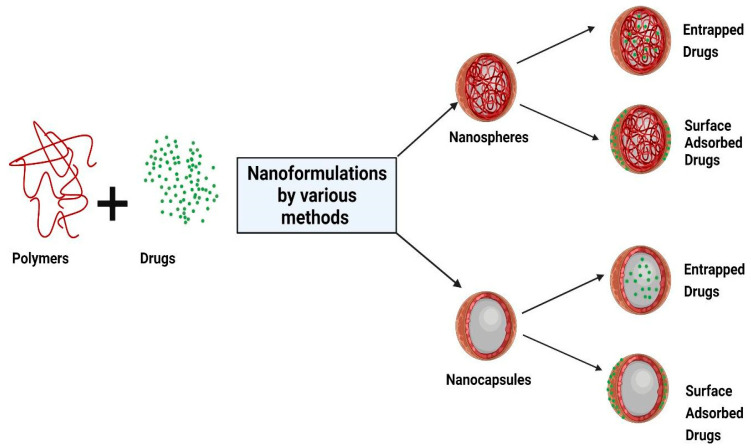
Schematic representation of the structure of nanocapsules and nanospheres. Created with BioRender.com (accessed on 23 December 2021).

**Figure 2 pharmaceutics-14-00872-f002:**
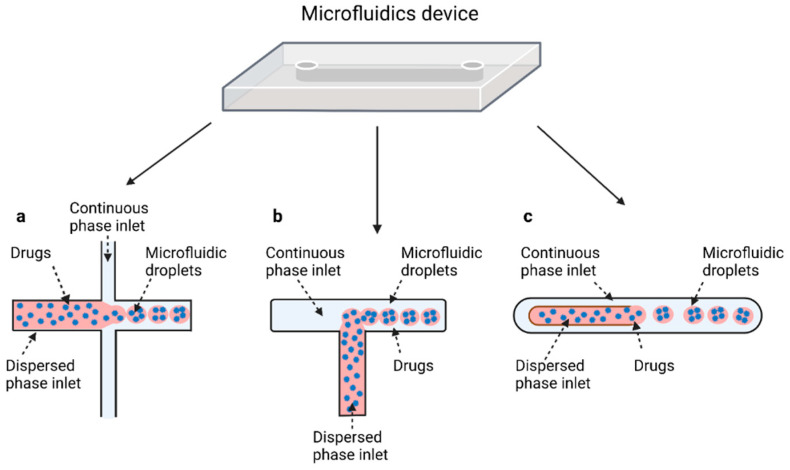
Schematic illustration of the most common designs used to make droplets with microfluidic devices. Flow-focusing (**a**): the two phases meet at the junction where the continuous phase pinches off droplets of the dispersed phase from the bottom and the top. T-junction (**b**), where both phases meet at a T-shaped junction and the shear stress created by the continuous phase causes droplet formation. (**c**) Droplet formation using a coaxial geometry, where the continuous phases completely surround the dispersed phase in three dimensions to create droplets. Created with BioRender.com (accessed on 24 December 2021).

**Figure 4 pharmaceutics-14-00872-f004:**
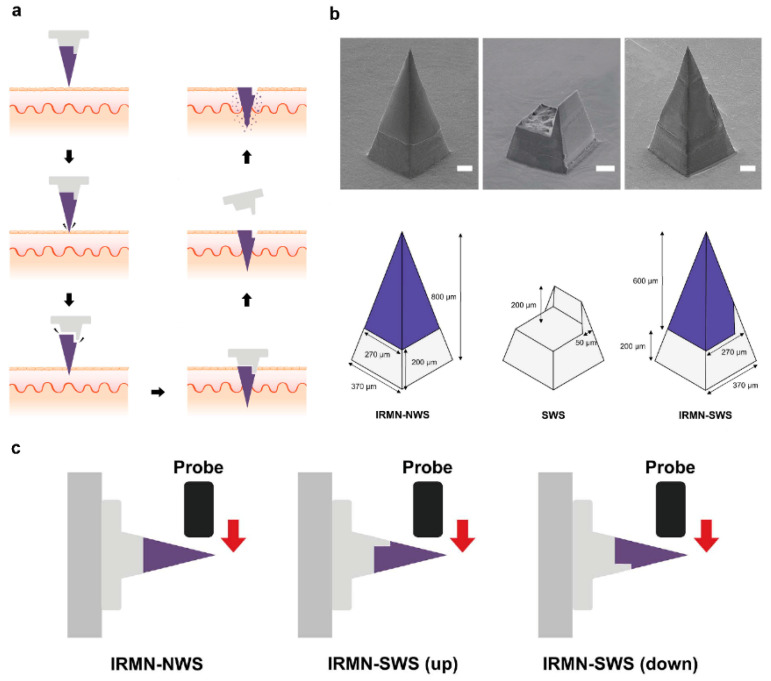
(**a**) Schematic illustration of insertion-responsive microneedles. (**b**) Describing the dimensions of different structures. (**c**) Schematic illustration of the transverse compression test. Reprinted from [108] 2018, Royal society of chemistry.

**Figure 5 pharmaceutics-14-00872-f005:**
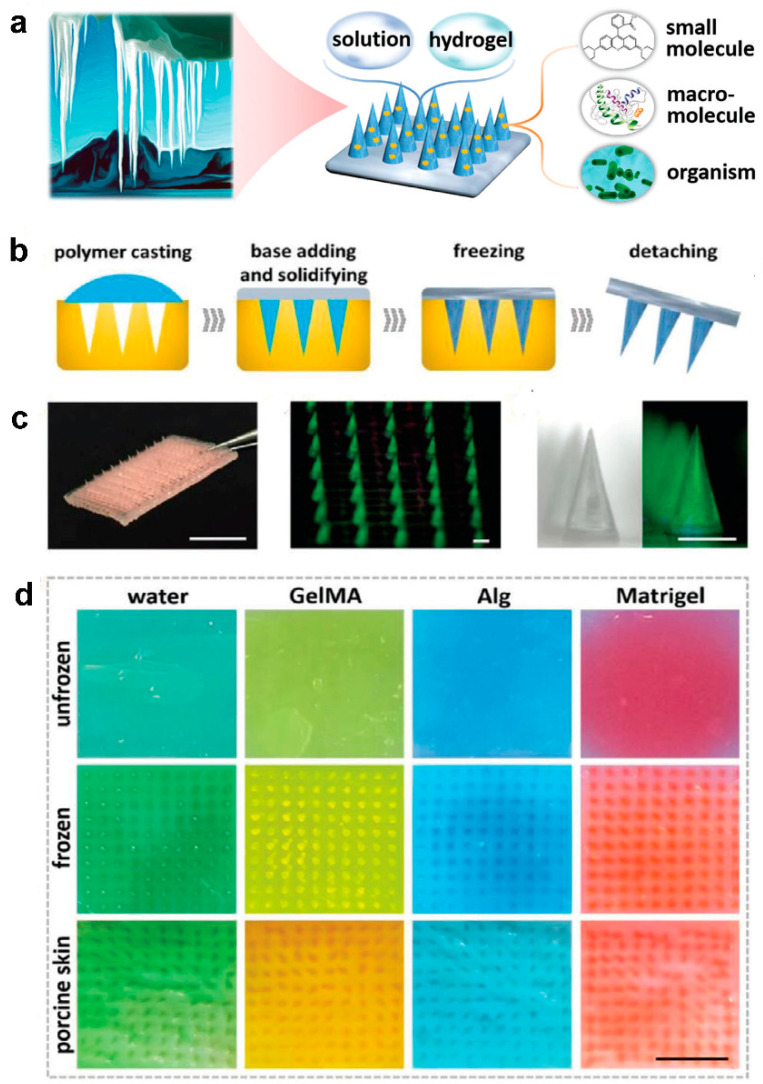
(**a**) Schematic illustrations of compositions and properties of ice microneedles. (**b**) Schemes of the fabrication process of ice microneedles. (**c**) Image of an ice microneedle patch. (**d**) Digital images of agarose after penetration of unfrozen water microneedles (dyed green), GelMA microneedles (dyed yellow), Alg microneedles (dyed blue), and Matrigel microneedles (dyed red); digital images of agarose after penetration of ice microneedles; digital images of porcine skins after penetration of ice microneedles. Reprinted from [112], 2020, Wiley.

**Figure 6 pharmaceutics-14-00872-f006:**
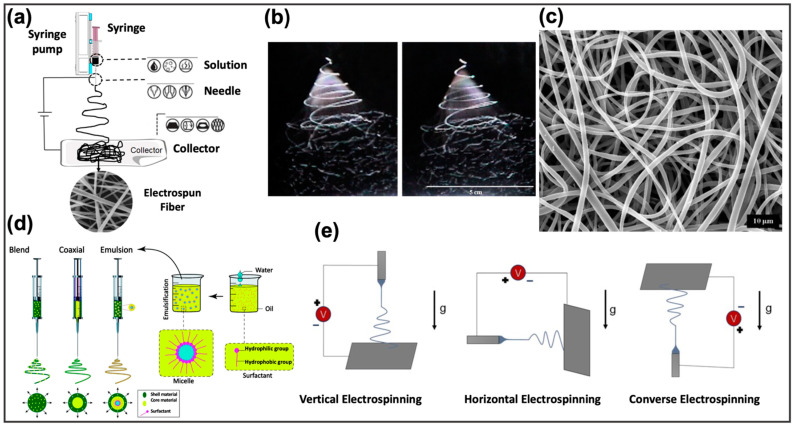
(**a**) Schematic representation of a classical electrospinning system. Reprinted from Fadil et al. [119], with permission under Polymers license. (**b**) Stroboscopic image of a polymeric jet, generated during electrospinning. Reprinted from [120] with permission under Polymer. (**c**) SEM image of the generated polymeric nanofibers. Reprinted from [120] with permission under Polymer. (**d**) Representation of the resulting nanofiber structures from different polymer feed solutions loaded for the electrospinning. Reprinted from Nikmaram et al. [121] with permission under RSC license. (**e**) Schematic representation of the impact of gravity on the resulting nanofibers in different orientations of the electrospinning system. Adapted from Nagam et al. [122] with permission under Fibers license.

**Figure 7 pharmaceutics-14-00872-f007:**
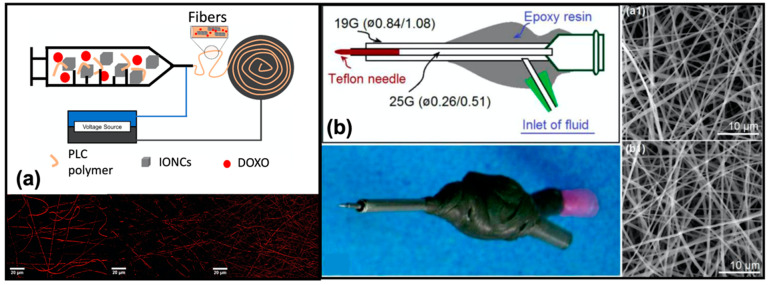
(**a**) Schematic representation of the system employed for fabrication of the magnetic fibers loaded with iron oxide nanocubes (IONCs) and DOXO, and confocal images of nanofibers with (15–23 nm) and without nanoparticles. Reprinted from [137] with permission under Journal of colloid and interface science license. (**b**) Schematic representation of the new proposed spinneret, with its digital photograph and SEM images of the obtained nanofibers from (**a1**) the traditional process (N1), and (**b1**) new ones (N2). Reprinted from [138].

**Figure 8 pharmaceutics-14-00872-f008:**
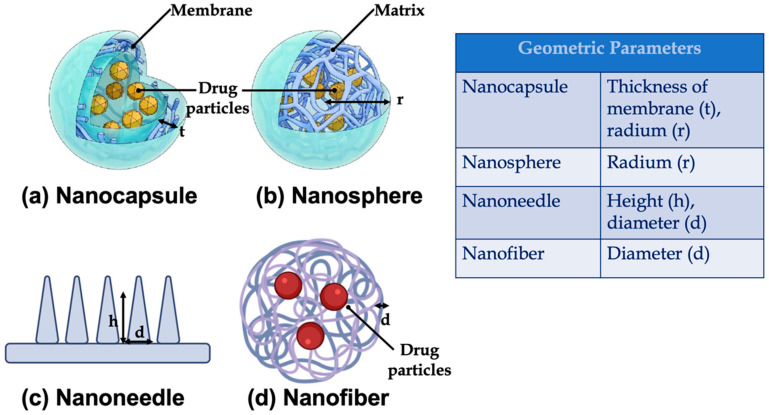
Main geometric parameters for (**a**) nanocapsule (reprinted from [150]), (**b**) nanosphere (reprinted from [150]), (**c**) nanoneedle (created with BioRender.com, accessed on 6 February 2022), and (**d**) nanofiber (created with BioRender.com, accessed on 6 February 2022).

**Figure 9 pharmaceutics-14-00872-f009:**
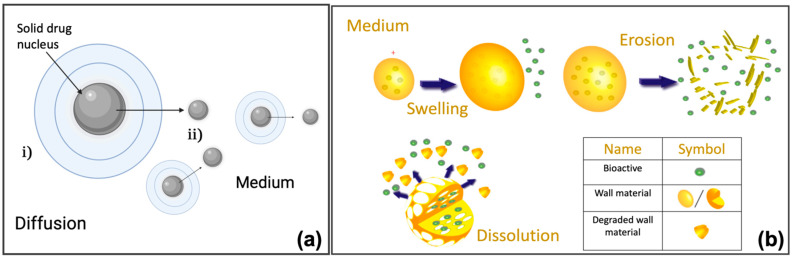
(**a**) Schematic representation of the drug diffusion processes from micro or nanoparticles. The figure shows a typical structure of micro/nanoparticles, with a solid drug nucleus: as the dissolution process proceeds, the solid drug diffuses from (i) the micro/nanoparticles to (ii) the medium. Created with BioRender.com (accessed 6 February 2022). (**b**) Representations of swelling, erosion, and dissolution. Reprinted from [159].

**Figure 10 pharmaceutics-14-00872-f010:**
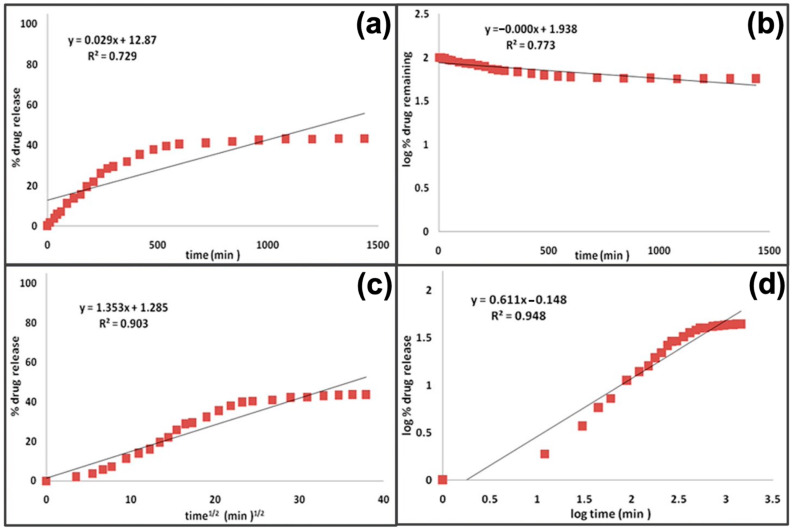
Comparison of different profiles of drug release kinetics: (**a**) Zero-order kinetic, (**b**) first-order kinetic, (**c**) Higuchi model, and (**d**) Korsmeyer–Peppas model. Reprinted from [161].

**Table 1 pharmaceutics-14-00872-t001:** Overview of encapsulation methods for the preparation of drug loaded nano/microcarriers.

Morphology	Polymer	Payload	Method	Indication	References
Nanostructure	chitosan layered	coenzyme Q10	oil in water nanoemulsions	anti-inflammatory effects	[44]
Nanostructure	chitosan layered	lycopene	oil in water nanoemulsions	cardioprotective effects	[46]
Nanostructure	chitosan layered	curcumin	oil in water nanoemulsions	anti-inflammatory	[48]
Nanostructure	PLGA	noscapine	oil in water nanoemulsions	anticancer	[65]
Nanostructure	PLGA	turmeric	emulsion-solvent evaporation	antioxidant activity	[66]
Nanostructure	polyallylamine	clindamycin	salting out	osteomyelitis treatment	[82]
Nanostructure	mPEG–PTMC	dexamethasone	salting out/double emulsion	anti-inflammatory	[83]
Nanostructure	PLA	Savoxepine	salting-out	neuroleptics effect	[84]
Nanostructure	PEG-*b*-PCL	aza-BODIPY	nanoprecipitation	cancer treatment	[85]
Microstructure	PLGA-PEG	clofazimine	nanoprecipitation	tuberculosis	[86]
Microstructure	alginate	probiotic *L. paracasei CBA L74*	water-in-oil emulsion technique	microbiota disfunction	[8]
Microstructure	PLGA	collagenase	water in oil in water	skin pathologies	[25]
Microstructure	PLGA	donepezil	oil in water emulsion	Alzheimer treatment	[64]
Microstructure	PLGA	vascular endothelial growth factor	water in oil in water	angiogenic effect	[26]
Microstructure	PLGA	laccase	water in oil in water	Cosmetic	[27]

Abbreviation: poly(ethylene glycol)-*block*-poly(ε-caprolactone)—PEG-*b*-PCL, poly(lactic-co-glycolic acid) and poly(ethylene glycol)—PLGA-PEG, poly(DL-lactic acid)—PLA, poly(trimethylene carbonate) and monomethoxy poly(ethylene glycol)-*block*-poly(trimethylene carbonate)—mPEG–PTMC.

**Table 2 pharmaceutics-14-00872-t002:** Flow/MP size/polymer concentration relationship. (https://microtas2020.org/images/current_support_files/dolomite/Dolomite_Application_Note-Microfluidic_Production_of_20_to_50_%C2%B5m_Core-Shell_PLGA_Beads.pdf accessed on 30 September 2019).

PLGA wt %	PLGA Flow Rate * μL/min	AP Flow Rate μL/min	Outer Droplet Size μm	Picture **	Final Bead Size μm	Picture **
2	5 (11.6)	70	57.2	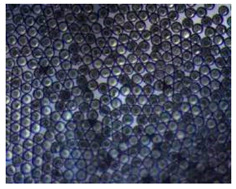	28.7	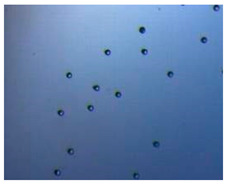
	2 (2.6)		57.7	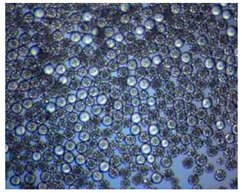	25.6	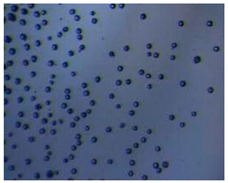
	2 (2.6)	30	67.4	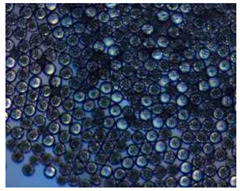	29.9	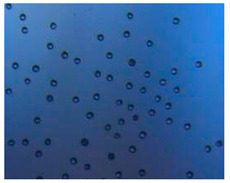
6	5 (10.1)	70	61.6	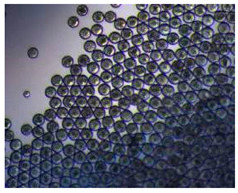	36.9	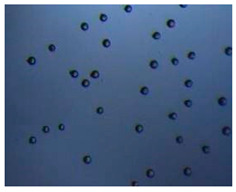
	2 (2.3)		58.4	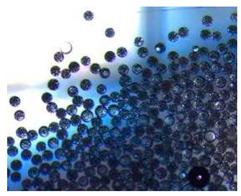	36.7	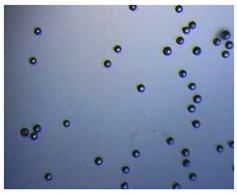
	2 (2.3)	30	74.9	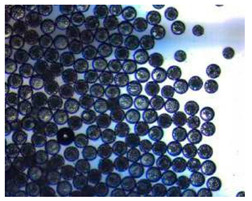	45.4	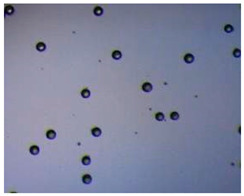
	5 (10.1)	30	72.2	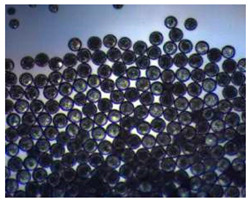	45.4	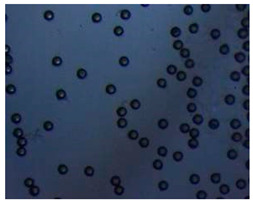

* Actual flow rates are shown within brackets and are calculated from set flow rate using the calibration curves reported in the appendix. ** The core-shell PLGA droplets are collected on a microscope slide before and after shrinkage. The particles look highly monodispersed with the water droplets clearly visible within.

**Table 3 pharmaceutics-14-00872-t003:** FDA-approved polymer-based drug delivery systems described in this review.

Type	Trade Name	Formulation	Indication
Polymeric Nanoparticles	Lupron Depot	microspheres composed of PLA-PLGA copolymer and leuprolide acetate	prostate cancer
	Sculptra	PLA microparticles	facial lipoatrophy, facial wrinkles
	Zilretta	triamcinolone acetonide embedded in a PLGA hydrogel	knee osteoarthritis
	Bydureon BCise	exenatide sustained-release	type 2 diabetes
	Adagen	PEGylated adenosine deaminase enzyme	immunodeficiency disease
	Cimzia	PEGylated antibody fragment	rheumatoid/psoriatic arthritis
	Neulasta	PEGylated form of filgrastim	Neutropenia
	Plegridy	PEG-IFN-β-1a	multiple sclerosis
Gelling systems	Eligard	leuprolide acetate and polymer PLGA	prostate cancer
	Zoladex	goserelin acetate dispersed in a cylindrical PLGA matrix	prostate cancer and endometriosis
	OncoGel	paclitaxel and a PLGA-PEG thermosensitive polymer system	solid tumors
Implants	Retisert	fluocinolone acetonide in a tablet reservoir	noninfectious uveitis
	Vitrasert	ganciclovir tablet reservoir	cytomegalovirus retinitis
	Ozurdex	dexamethasone sustained-release implant	macular edema and noninfectious uveitis
Polymeric Microneedles	V-Go	TOPAS cyclic olefin copolymer	type 2 diabetes
	Sanofi’s Fluzone Intradermal Quadrivalent	n.d.	influenza A subtype viruses and type B viruses
Electrospinning based nanofibers	Rivelin	muco-adhesive two layered patch	mucosal diseases
	SurgiClot	fibrin sealant patch	bone bleeding

## Data Availability

https://microtas2020.org/images/current_support_files/dolomite/Dolomite_Application_Note-Microfluidic_Production_of_20_to_50_%C2%B5m_Core-Shell_PLGA_Beads.pdf (accessed on 24 February 2022), https://stteresamedical.com/technology/surgiclot (accessed on 24 February 2022).

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
