# Peer review of "Recent Fabrication Methods to Produce Polymer-Based Drug Delivery Matrices (Experimental and In Silico Approaches)"

_pharmaceutics, 2022, doi:10.3390/pharmaceutics14040872_

Round 1

Reviewer 1 Report

In the manuscript entitled: “Experimental and in Silico approaches to produce Polymer-Based Drug Delivery Matrices”, the authors described experimental studies of various drug delivery devices, biodegradable polymeric-based systems in the micro- and nanometer range. Together with silico approaches, the drug release of these systems can be optimized, which allows to further reduce the time and cost associated with drug development. Overall, the manuscript is well written. It is suitable for readers of Pharmaceutics, as it addresses an important issue of the combination of experimental and in silico approaches to help in designing nanoparticles for drug delivery, and I recommend the manuscript to be accepted. However, I recommended the authors to include more detailed description of using in silico methods for designing the polymeric-based drug delivery system. The authors have a comprehensive demonstration for the experimental research on biopolymers, however the content for in silicon methods was clear not enough. It will be great if the authors can provide more evidences showing how in silico methods can help the rational design for polymer-based drug delivery system, not only focus on the mechanism of drug release. Here are some directions for you to consider. - computer simulations in designing nanoparticles for drug delivery - in-silico screening for drug solubility (solubility modeling) - application of quality by design (QbD) in drug development (design of experiments, DOE) - molecular simulations for formulation development

Reviewer 2 Report

Procopio et al., has reviewed experimental and mathematical modeling approaches to the development of drug delivery system with special focus on the polymers. In general, it is well-written. A few suggestions need to be addressed before it can be published,

  1. The title needs to be more focused. The manuscript has 50% space for fabrication of drug delivery system, which is not reflected in the title. The same is applied to the conclusion.
  2. In the section of “micro- and nano-particles”, authors discussed the effects of physical and chemical properties of particles on the drug release/encapsulation, efficacy and degradation. But “technology” in terms of particle synthesis and preparation is absent.
  3. The section of 2.2 reviewed the layered nano-emulsion, which belongs to polymer-lipid hybrid nanoparticles (PLN). Authors need to briefly the advantages of PLN in general.
  4. In section 2.3, a Table of comparing different fabrication technique would be helpful.
  5. In microfluidics section (2.4), the flow rate and oil/aqueous phase are important parameters to determine the particle properties. It should be discussed and provide the examples here.
  6. In the section of “silico models”, drug release kinetics determined by mechanistic and mathematical strategies are discussed. However, the drug release kinetics are also influenced by the drug release method, such as dialysis, centrifugation, which also needs to be mentioned.
  7. Minor edits: “thanks to” is better to be replace with “owing to” or “due to”; In Line 119 – 121), “as well as to microparticles…” may be placed in the sentence, rather than in the beginning of paragraph.

Round 2

Reviewer 2 Report

the manuscript has been significantly improved.